# Targeting the Metabolic Paradigms in Cancer and Diabetes

**DOI:** 10.3390/biomedicines12010211

**Published:** 2024-01-17

**Authors:** Mira Bosso, Dania Haddad, Ashraf Al Madhoun, Fahd Al-Mulla

**Affiliations:** 1Department of Pathology, Faculty of Medicine, Health Science Center, Kuwait University, Safat 13110, Kuwait; 2Department of Genetics and Bioinformatics, Dasman Diabetes Institute, Dasman 15462, Kuwait; dania.haddad@dasmaninstitute.org (D.H.); ashraf.madhoun@dasmaninstitute.org (A.A.M.); 3Department of Animal and Imaging Core Facilities, Dasman Diabetes Institute, Dasman 15462, Kuwait

**Keywords:** oxidative phosphorylation, cancer, mitochondria, metabolic shift, type 2 diabetes, insulin resistance, therapy, nutritional adjuvants, glutaminolysis

## Abstract

Dysregulated metabolic dynamics are evident in both cancer and diabetes, with metabolic alterations representing a facet of the myriad changes observed in these conditions. This review delves into the commonalities in metabolism between cancer and type 2 diabetes (T2D), focusing specifically on the contrasting roles of oxidative phosphorylation (OXPHOS) and glycolysis as primary energy-generating pathways within cells. Building on earlier research, we explore how a shift towards one pathway over the other serves as a foundational aspect in the development of cancer and T2D. Unlike previous reviews, we posit that this shift may occur in seemingly opposing yet complementary directions, akin to the Yin and Yang concept. These metabolic fluctuations reveal an intricate network of underlying defective signaling pathways, orchestrating the pathogenesis and progression of each disease. The Warburg phenomenon, characterized by the prevalence of aerobic glycolysis over minimal to no OXPHOS, emerges as the predominant metabolic phenotype in cancer. Conversely, in T2D, the prevailing metabolic paradigm has traditionally been perceived in terms of discrete irregularities rather than an OXPHOS-to-glycolysis shift. Throughout T2D pathogenesis, OXPHOS remains consistently heightened due to chronic hyperglycemia or hyperinsulinemia. In advanced insulin resistance and T2D, the metabolic landscape becomes more complex, featuring differential tissue-specific alterations that affect OXPHOS. Recent findings suggest that addressing the metabolic imbalance in both cancer and diabetes could offer an effective treatment strategy. Numerous pharmaceutical and nutritional modalities exhibiting therapeutic effects in both conditions ultimately modulate the OXPHOS–glycolysis axis. Noteworthy nutritional adjuncts, such as alpha-lipoic acid, flavonoids, and glutamine, demonstrate the ability to reprogram metabolism, exerting anti-tumor and anti-diabetic effects. Similarly, pharmacological agents like metformin exhibit therapeutic efficacy in both T2D and cancer. This review discusses the molecular mechanisms underlying these metabolic shifts and explores promising therapeutic strategies aimed at reversing the metabolic imbalance in both disease scenarios.

## 1. Introduction

Cellular metabolism involves a series of enzyme-driven biochemical reactions that generate or consume energy. The activity and speed of these reactions fluctuate constantly. Diverse cellular energy requirements, proliferative activities, environmental stressors, and overall functions govern these fluctuations. Nevertheless, metabolism is now perceived in much broader ways than mere biochemistry; it permeates all facets of biology [1,2].

Under healthy conditions, cells can balance anabolism, catabolism, and waste removal by monitoring and coordinating different metabolic pathways. In various disease states, this intricate balance is lost, resulting in altered metabolism. Usually, genetic reprogramming underlies dysfunctional metabolic switching in cells and tissues. These perturbing shifts in metabolism are different in each disease. Here, we highlight the paradox of the metabolic shift in cancer versus type 2 diabetes (T2D) and its implications in targeted therapy [1,2]. The underlying metabolic pathways for both cancer and T2D continue to be examined. Notably, a higher incidence of cancer was observed in diabetic patients than in non-diabetic patients [3]. Many underlying cell signaling pathways for both conditions do intersect. In this review, we highlight important ones involved in the shift from oxidative phosphorylation (OXPHOS) to glycolysis and vice versa.

In 1923, Otto Warburg first postulated the Warburg phenomenon, suggesting metabolic rewiring to be one of the hallmarks of cancer, after observing that tumors demonstrate increased glucose uptake. He further hypothesized that cancer cells, due to dysfunctional mitochondria, primarily utilize aerobic glycolysis instead of OXPHOS for rapid energy release, which is required by proliferating cells [4,5]. The Warburg effect involves cytoplasmic anaerobic fermentation of glucose into lactate, despite regular oxygen availability. Aerobic glycolysis ultimately increases cellular anabolism and decreases catabolism. Our knowledge of this phenomenon and its driving forces has been refined and expanded over the past decades. Nevertheless, two features of the Warburg effect remain unaltered: increased glucose uptake and lactate production [6,7]. Aerobic glycolysis is markedly heightened in over 70% of cancer types, such as lung [8], breast [9], liver [10,11], brain [12], prostate [13], gynecologic [14], and pancreatic cancer [4,15,16]. Similar to solid tumors, hematologic malignancies, such as lymphomas [16,17,18] and leukemias [16,19,20], also demonstrate high aerobic glycolysis and low OXPHOS rates. In certain tumors, the accelerated Warburg effect occurs even in the presence of active or partially active mitochondrial OXPHOS [21,22,23,24]. It is argued that, in cancer, minimal activity of mitochondrial OXPHOS is crucial for tumor cell survival [22,25,26,27] and metastasis [28]. As discussed in further sections, accumulating evidence suggests additional reasons why mitochondrial dysfunction plays a role in the preferential glycolytic shift in tumor cells.

In addition to the Warburg effect, cancer cells simultaneously adopt another metabolic pathway called glutaminolysis as part of a metabolic reprogramming strategy to meet their specific energy and biosynthetic demands. With heightened energy demands, cancer cells absorb and utilize more glutamine than normal cells, supplementing glucose as an additional energy source [29]. Glutaminase converts glutamine to glutamate and ammonia [30]. The resultant glutamate enters the tricarboxylic acid (TCA) cycle within mitochondria, supporting energy production and biosynthetic precursor synthesis [31]. Notably, increased glutaminolysis in cancer cell mitochondria induces a metabolic shift from canonical OXPHOS and ATP production to the synthesis of anabolic intermediates for lipid and amino acid production. Malate, an intermediate in the TCA cycle, is metabolized to pyruvate and lactate, whereas citrate contributes to lipid metabolism. Both processes generate NADPH molecules, countering specific reactive oxygen species and averting oxidative stress [32]. Furthermore, glutamine serves as a crucial nitrogen source for nucleotide biosynthesis in the cytosol. In purine biosynthesis, two glutamine molecules provide nitrogen atoms for the purine ring formation in inosine monophosphate, a precursor to both adenosine monophosphate and guanosine monophosphate [33]. In pyrimidine biosynthesis, one glutamine molecule provides the nitrogen atom necessary for the formation of cytidine triphosphate from uridine triphosphate [34]. Collectively, glutamine acts as a signaling molecule, activating essential pathways that promote survival, proliferation, and differentiation.

Notably, a pivotal 2020 study by K.I. Nakayama found a significant shift in the fate of glutamine-derived nitrogen in cancer, which is crucial for cell proliferation and survival [35]. The nitrogen’s fate shifts away from the anaplerotic pathway supporting the TCA cycle, redirecting towards nucleotide biosynthesis. The regulation of this shift lies in the enzymes glutaminase (GLS1) and phosphoribosyl pyrophosphate amidotransferase (PPAT) [35]. A higher PPAT/GLS1 ratio orchestrates this transition, with PPAT steering nitrogen metabolism towards nucleotide synthesis and reduced GLS1 expression. GLS1, on the other hand, guides nitrogen metabolism to produce glutamate and ammonia, a pivotal step influencing the TCA cycle. Heightened GLS1 activity hinders tumor growth, whereas increased PPAT activity supports cell proliferation. Consequently, the determining factor for the metabolic shift is not solely glutamine availability but rather the PPAT/GLS1 ratio, as emphasized in the study [35]. In cancer, a prevalent pattern is observed with elevated PPAT expression and diminished GLS1 expression, particularly during malignant transformation [35]

In certain cancers, dysregulation of these pathways may contribute to cancer development and metastasis.

In contrast, the elaborate metabolic alterations characterizing T2D diverge from those associated with cancer. Despite these variances, there are shared signaling molecules at the crossroads of both conditions that influence shifts in OXPHOS and/or glycolysis. 

In established T2D, insulin resistance arises in peripheral tissues, primarily in the skeletal muscle (SKM) [36], and decreases glucose-induced insulin secretion by pancreatic β cells [37,38,39,40]. In healthy SKM, insulin increases the mitochondrial capacity for OXPHOS via an increased expression of mitochondrial OXPHOS-related genes and the posttranslational modification of mitochondrial proteins in the form of phosphorylation [41,42,43,44]. Hyperglycemia induces the release of insulin, activating mitochondrial respiration [45]. However, chronic hyperglycemia in individuals, due to continuous nutritional overload and decreased physical activity, leads to prolonged hyperactivity of the OXPHOS machinery. This is associated with a consistently excessive release of reactive oxygen species (ROS), leading to oxidative toxicity and insulin resistance in peripheral tissues, which eventually results in the development of T2D [46,47,48]. However, in established T2D, contradictory findings have been reported on mitochondrial OXPHOS in SKM [43]. Several researchers reported mitochondrial dysfunction and low OXPHOS in SKM [49,50], whereas others reported normal OXPHOS [51,52,53]. A few studies reported that the liver exhibited normal to even increased mitochondrial OXPHOS [54,55,56]. Some researchers argue that mitochondrial dysfunction contributes to the development of insulin resistance and T2D [49,50,57]. However, the opposite is more often believed to be true, i.e., insulin resistance leads to mitochondrial dysfunction in peripheral tissues [43,58]. In contrast, in T2D, a Warburg-like effect and lactate production also occur in pancreatic β cells [59,60]. The released lactate could cause insulin resistance by suppressing glycolysis and impairing insulin signaling in SKM [61].

In this review, we delve into the main metabolic patterns associated with cancer and T2D, shedding light on several encouraging nutritional and therapeutic methods. These approaches are intended to counteract the metabolic changes, working towards reinstating a typical balance in both diseases.

## 2. The Metabolic Shift in Cancer

### 2.1. Why Do Tumors Adopt Glycolysis over OXPHOS?

Both anaerobic glycolysis and OXPHOS produce cellular energy in the form of adenosine triphosphate (ATP). OXPHOS is much more efficient in generating ATP than glycolysis; it generates approximately 32 ATPs from a single glucose molecule, whereas glycolysis produces only a net of two ATPs (see Figure 1). Briefly, during glycolysis, a single molecule of glucose is converted into two molecules of pyruvate through a series of biochemical reactions that ultimately result in the production of two ATPs, after consuming two ATPs in the glycolytic process. The resultant pyruvate can either enter the Krebs cycle (tricarboxylic acid (TCA) cycle) followed by OXPHOS in the mitochondria in aerobic conditions or it can be converted to lactate in anaerobic conditions. OXPHOS is an oxygen-dependent process that combines the oxidation of nicotinamide adenine dinucleotide (NADH) and flavin adenine dinucleotide (FADH_2_) with the phosphorylation of ADP to form ATP. In contrast, in anaerobic glycolysis, the conversion of pyruvate to lactate consumes NAD^+^ to generate NADH. The biochemical landscapes of glycolysis, aerobic glycolysis, and the TCA cycle were well reviewed by Akram in 2013 [62,63]. Another recent review by Shiva et al. (2020) elegantly describes OXPHOS, which is also known as the electron transport chain (ETC) cycle [64]. The selection of a less efficient metabolic pathway by the cell is attributed to the fast-track generation of ATP by glycolysis when compared to OXPHOS. Therefore, it was believed that an increase in the rate of aerobic glycolysis would promptly deliver the energy needs of the cell [23,65,66,67,68]. In that sense, an increase in the frequency of aerobic glycolysis translates into an increase in glucose uptake [69] and rapid ATP synthesis. However, this explanation was found to be insufficient to justify the glycolytic surge in tumors even in the presence of functional mitochondria. Other reasons and theories have emerged, discussed in the following sections (see Figure 2). Notably, another theory of altered energy metabolism in tumor cells exists and is termed the reverse Warburg effect or tumor symbiosis. This theory emphasizes crosstalk between hypoxic tumor cells and normoxic stromal cells. Stromal cells uptake the lactate generated by hypoxic tumor cells, using it as fuel to generate ATP via oxidative mechanisms. The ATP generated by normoxic cells then becomes a source of energy for neighboring hypoxic tumor cells [70]. This lactate shuttle and the interplay between the heterogeneous metabolic phenotypes in the core and microenvironment of the tumor contribute to the survival of tumor cells [70,71,72,73].

In the next subsections, we summarize the reasons behind the adoption of glycolysis over OXPHOS in tumor cells.

#### 2.1.1. Reason 1: Mitochondrial Dysfunction

Despite controversies around the presence of functional or partially functional mitochondria in some cancers, we believe these observations are an exception rather than the standard. Discovering signs of mitochondrial activity in some cancer phenotypes does not equate to the presence of normally functional mitochondria. Mitochondrial dysfunction continues to be the leading cause of predominantly occurring glycolysis over OXPHOS in tumor cells. Several mechanisms contribute to the development of mitochondrial dysfunction in tumor cells (see Figure 3); this is fortified by the mutations or transcriptomic dysregulations found in the genes that encode OXPHOS- and glycolysis-related proteins (Table 1 and Table 2).

OXPHOS-related genes are either of mitochondrial DNA (mtDNA) or nuclear DNA (ncDNA) origin. Point mutations are the most common type of mutations in mtDNA reported to date. Particularly interesting are the mutations occurring in oncogenes and tumor suppressor genes, which are frequent and characteristic of all cancer types. Many oncogenes and tumor suppressor genes indirectly regulate glycolysis and OXPHOS by regulating the expression of OXPHOS-related or glycolytic proteins (Table 3). Moreover, mitochondrial metabolic dysfunction in cancer can also result from the dysregulation of mitochondrial biogenesis and mitophagy. In the context of cancer, mutations in ncDNA genes can affect the expression or activity of TCA cycle-related enzymes, such as succinate dehydrogenase (SDH) [74], fumarate hydratase (FH) [75], and isocitrate dehydrogenase (IDH) [76,77], which have a direct effect on OXPHOS [78] (Table 1). For instance, Bourgeron et al. reported—for the first time—that a mutation in the SDH gene caused mitochondrial ETC deficiency [79]. In 2018, Böttcher et al. discovered that a gain-of-function mutation in IDH leads to enhanced D-2HG production, which triggers the destabilization of the hypoxia-inducible factor-1 alpha (HIF-1α) protein, thus making the cell more dependent on OXPHOS [80].

Likewise, tumorigenic mutations in cardinal oncogenes and tumor suppressor genes significantly contribute to mitochondrial dysfunction in cancer. Although oncogenes and tumor suppressor genes do not directly encode OXPHOS-related or glycolytic proteins, they indirectly regulate the activity of these proteins through cell signaling pathways (Table 2). This strongly abates the rationale for asserting the presence of functional mitochondrial OXPHOS in cancer. Common oncogenes and tumor suppressor genes involved in the mitogen-activated protein kinase (MAPK), phosphoinositide 3-kinase (PI3K), and mammalian target of rapamycin (mTOR) pathways have been heavily reported in most cancer types and can affect mitochondrial function (Table 2).

Mitochondrial deregulation is also manifested by an imbalance in the degree of biogenesis of mitochondrial organelles and mitophagy of unhealthy mitochondria. Imbalances in mitochondrial organelle turnover result in an abnormal number of available mitochondrial organelles in the cytoplasm, which have been implicated in cancer progression [81,82]. Autophagy is inhibited by mTOR pathway activity. An active AMP-activated protein kinase (AMPK) pathway inhibits the mTOR pathway, thereby activating autophagy. Autophagy promotes cell survival by recycling cellular organelles to produce energy. Reportedly, there is an association between high autophagic activity and increased cancer resistance to chemotherapy [83,84]. Mitophagy, on the other hand, degrades mitochondria through either the PTEN-induced kinase 1 (PINK1)/Parkin (PRKN) or the BCL2 interacting protein 3 (BNIP3)/NIP-3-like protein X (NIX)/FUN14 domain-containing 1 (FUNDC1) pathways or by AMPK activation and consequent phosphorylation of Unc-51-like autophagy activating kinase 1 (ULK1). Interestingly, some studies have shown that the downregulation of PRKN causes a decrease in mitophagy and the accumulation of dysfunctional mitochondria in the cytoplasm. This has been associated with decreased mitochondrial OXPHOS, increased ROS, and increased glycolysis. Therefore, PRKN deficiency contributes to the Warburg effect in cancer. PRKN deficiency has been observed in several cancer phenotypes in humans, including colorectal cancer [85], glioblastoma [86], melanoma [87], lung cancer [88], and breast cancer [81,89].

Some studies have suggested that the overexpression of uncoupling protein (UCP) promotes aerobic glycolysis, tumor proliferation, and resistance to apoptosis-induced chemotherapy [90,91,92]. UCPs are a family of mitochondrial proteins localized in the mitochondrial membrane that act as anion transporters. UCP2 in particular is ubiquitously expressed in the body and plays several biological functions and has been shown to play a role in both tumorigenesis and chemoresistance. UCP2 has an antioxidant effect due to its role in transporting protons from the inner mitochondrial membrane to the inner mitochondrial matrix [93]. In 2016, Brandi et al. demonstrated that UCP2 caused the downregulation of OXPHOS-related complex I (NADH dehydrogenase), complex IV (cytochrome c oxidase), and complex V (ATPase) and a decrease in mitochondrial oxygen consumption [92].

**Table 1 biomedicines-12-00211-t001:** Common alterations in the genes involved in the TCA cycle and OXPHOS metabolism in cancer.

Gene	Encoding DNA	Protein	Cycle	Reported Dysregulation in Cancer	Publications
*Aco2*	Nuclear	Aconitase 2	TCA (Krebs cycle)	Overexpression Increased activity	[78,94]
*IDH1*	Nuclear	Isocitrate Dehydrogenase 1	TCA	Point mutations	[76,78,95,96,97,98,99]
*SDH*	Nuclear	Succinate Dehydrogenase	TCA and ETC cycles	Inherited or somatic mutations in the SDH Downregulation	[74,78,100,101,102,103,104,105,106,107,108]
*FH*	Nuclear	Fumarate Hydratase	TCA	Germline mutations Reduced FH gene expression	[75,78,109,110,111,112,113]

**Table 2 biomedicines-12-00211-t002:** Common oncogenes and tumor suppressor genes implicated in mitochondrial dysfunction in cancer.

Gene	Class	Genetic Alteration	Pathway Affected	Effects on OXPHOS (ETC Cycle)	Effect of Cancer Progression	References
*MYC* (MYC proto-oncogene protein)	Oncogene	Point mutation, amplification	TGF-β signaling pathway	Stimulates mitochondrial biogenesis and function through regulating the transcription factor A mitochondrial gene	Self-sufficiency in growth status	[78,114,115,116]
*AKT* (alpha serine/threonine kinase)	Oncogene	Point mutation, amplification, overexpression	AKT pathway	Affects mitochondria membrane potential (DWmt). Activated PI3-K–AKT pathway enhances mitochondrial membrane stability by inhibition of p53 and Bax expression to limit mitochondria-associated apoptosis. Stimulates the glycolysis pathway. PTEN inactivation upregulates mitochondrial respiratory capacity through the 4E-BP1-mediated protein translation pathway.	Evade apoptosis	[115,117,118]
*P53*	Tumor suppressor gene	Point mutation, deletion	P53 pathway, cell cycle control: G2/M DNA damage checkpoint	P53 downregulation blocks its transcriptional activity and its localization to mitochondria, thus inhibiting mitochondrial-mediated apoptosis and enhancing mitochondrial DNA (mtDNA) mutagenesis. P53 downregulation reduces SCO2 gene expression and cytochrome-c to molecular oxygen, thus maintaining the proton gradient across the inner mitochondrial membrane that is necessary for aerobic ATP production.	Evade apoptosis, insensitivity to anti-growth signals	[115,119,120]
*PI3K* (phophatidylinositol-4,5-bisphosphate 3-kinase)	Tumor suppressor	Point mutation	AKT pathway	Downregulation of PI3K activates and upregulates AKT signaling and mTOR downstream transcription of p70, which regulates the transcription of key apoptosis regulatory proteins. Decrease in mitochondrial membrane potential. Decrease in the release of cytochrome-c into the cytoplasm. Prevent activation of the proapoptotic caspase family of proteins does not get activated.	Evade apoptosis	[115,121]
*PTEN* (phosphatase and tensin homolog)	Tumor suppressor	Point mutation, deletion	PI3K pathway	PTEN downregulation activates PI3-K–AKT pathway. Decreased mitochondrial membrane stability via inhibition of the proapoptotic proteins p53 and Bax expression to limit mitochondria-associated apoptosis (intrinsic pathway).	Evade apoptosis	[115,122]
*MDM2* (mouse double minute 2, human homolog of; P53-binding protein)	Oncogenes	Amplification	Cell cycle control: G1/S checkpoint	Negatively regulates NADH: ubiquinone oxidoreductase, 75 kDa Fe-S protein 1 (NDUFS1), and NADH dehydrogenase 6 (MT-ND6) involve the d in the ETC cycle. MDM2 overexpression decreases the function and efficiency of mitochondrial complex I (CI).	Evade apoptosis	[115,123]
*BRAF* (B-Raf proto-oncogene, serine/threonine kinase)	Oncogenes	Point mutation, amplification, increased expression	MAPK pathway (RAS)	BRAF upregulation inhibits oxidative phosphorylation gene transcription, mitochondrial b, biogenesis, and the expression of PGC1a by targeting the melanocyte lineage factor (MITF).	Self-sufficiency in growth status	[115,124]
*KRAS* (Kirsten rat sarcoma viral oncogene homolog, GTPase)	Oncogene	Point mutation	MAPK pathway	KRAS activation of MAPK and PI3K pathways stabilizes and activates hypoxia-inducible factor-1 alpha and factor-2 alpha (HIF-1α and HIF-2, respectively), which facilitates ischemic adaptation. KRAS stimulates aerobic glycolysis by overexpressing hexokinase, lactate dehydrogenase, and glucose transporters. KRAS induces glutaminolysis by upregualting glutamate oxaloacetate transaminase 1,2 (GOT), leading to aspartate and NADPH generation and the activation of the NRF2 antioxidant system. Upregulation of RAS leads to increased autophagy and micropinocytosis, contributing to the disruption of cellular energy balance and nutrient scavenging.	Self-sufficiency in growth status	[115,125,126,127,128]
*NF-κB* (nuclear factor kappa B)	Oncogene	Amplification, rearrangement, chromosomal translocation in several members of the NF-κB protein family or constitutional activation of NF-κB	NF-κB pathway	NF-κB upregulation and activity cause a decline in mitochondrial respiratory capacity and reduce the expression of key mitochondrial proteins, including SDHA, ANT-1, UCP3, and MFN2, and cause increased fission and mitophagy of mitochondrial organelles. It upregulates PGC1α and correlates with high ROS.	Tumor growth	[115,129,130]
*EGFR* (ErbB1 epidermal growth factor receptor)	Oncogene	Amplification, upregulation	PI3K and MAPK pathways	EGFR modulates mitochondrial function through modification of Cox-II.	Self-sufficiency in growth status	[115,131]
*IGFR* (insulin-like growth factor receptor)	Oncogene	Amplification	AKT, PI3K, and MAPK pathways	Increased IGFR expression alters ATP synthesis, increases mitochondrial function, and decreases mitochondrial ROS production associated with the induction of antioxidant response.	Antiapoptotic, cell-survival, and transforming activities	[115,132]
*ErbB2* (HER2, receptor tyrosine protein kinase erbB-2 )	Oncogene	Amplification	MAPK, PI3K, AKT, and mTOR	ErbB2 overexpression causes downregulation of pro-apoptotic Bcl-2 family protein (Bcl-xS) and increases levels of anti-apoptotic Bcl-xL. This leads to mitochondrial dysfunction and a loss of mitochondrial membrane potential, a 35% decline in ATP levels, and a loss of redox capacity (mitochondrial reductase activity).	Anti-apoptotic and pro-proliferative effects	[115,133]
*HIF-1 α* (hypoxia inducible factor 1 subunit alpha)	Oncogene	It is stabilized and activated in hypoxic tumor conditions and by inactivating mutations of SDH, FH, and IDH as well as due to oncogenic mutation activating other signaling pathways (MAPK, AKT, and mTOR)		HIF-1α induces the expression of pyruvate dehydrogenase kinase 1 (PDK1). PDK1 phosphorylates and inactivates mitochondrial pyruvate dehydrogenase and enhances the dependence of cells on glycolysis for ATP production instead of OXPHOS.	Metabolism, cell survival, erythropoiesis, angiogenesis	[134,135,136]

#### 2.1.2. Reason 2: Glycolysis Supports the Proliferative Needs of Cancer Cells

In cancer, tumor cells employ strategies that promote their survival, growth, and invasion. Therefore, it has been theorized that cancer cells use aerobic glycolysis as a trade-off because it supports the biosynthetic anabolic needs of constant, uncontrolled proliferation [69].

The Warburg effect supplies nucleic acids, proteins, and lipids through certain branching pathways that emanate from glycolysis. For instance, the pentose phosphate pathway (PPP) generates the reducing agent NADPH, which is crucial for de novo lipid synthesis [137,138,139]. Moreover, the redirection of glycolysis flux towards de novo serine biosynthesis is facilitated by phosphoglycerate dehydrogenase (PHGDH) [69,140]. Additionally, lactate is produced during the final step of anaerobic glycolysis, along with NAD^+^. The produced NAD^+^ acts as a positive feedback mechanism, sustaining active glycolysis to ensure the continuous supply of building blocks [141]. Intracellularly produced lactate is transported to the extracellular stroma, contributing to its acidic attributes in cancer [142]. An insightful study by Heiden et al. revealed that the increased cellular demand for NAD+, surpassing the demand for ATP and the rate of ATP turnover, drives the preferential reliance on aerobic glycolysis, rather than OXPHOS, in proliferating cells such as cancer cells [143]. The NAD^+^/NADH ratio is critical for several metabolic processes, including nucleotide synthesis, lipid metabolism, amino acid metabolism, and central carbon metabolism [144]. Both redox reactions and biosynthetic processes necessitate NAD+ generation [144]. However, NAD+ regeneration by the ETC cycle is constrained due to increased mitochondrial membrane potential and decreased ATP synthase activity during OXPHOS in proliferating cancer cells [144]. To meet the heightened NAD+ demand, the cell diverts its metabolic phenotype towards aerobic glycolysis [144].

Undifferentiated stem cells resemble cancer cells in that they have high proliferative activity, and therefore, they similarly shift their metabolism towards anaerobic glycolysis instead of OXPHOS [145,146]. During stem cell differentiation, cellular metabolism switches back to mitochondrial OXPHOS to generate energy, and the rate of anaerobic glycolysis declines [147]. The dysregulation of the intracellular and extracellular pH of cancer cells that accompanies aerobic glycolysis is another means by which the Warburg effect promotes tumor growth [142]. Dysregulated pH dynamics characterized by extracellular acidic stromal cells and intracellular alkaline cytoplasm are hallmarks of cancer cells and can influence tumor proliferation, metastasis, and metabolic shift [142]. Proliferating cancer cells require an alkaline intracellular pH compared to normal quiescent cells that have an acidic intracellular pH [148,149,150]. The increase in intracellular pH in a cancer cell promotes the glycolytic metabolic shift and confers a proliferative advantage for tumor cells [151,152,153]. For instance, a cytoplasmic alkaline pH is required for growth factors to initiate nucleic acid synthesis [149]. Moreover, an alkaline intracellular pH promotes intracellular protein synthesis and drives other phenotypes of cancer cells [142,154,155].

#### 2.1.3. Reason 3: Activation of HIF-1α by ROS

Another reason for the glycolytic shift in cancer is the accumulation of ROS, which causes the activation of HIF-1α. Usually, in hypoxic conditions, HIF-1α gets activated when the cell senses a low oxygen supply. ROS mimics the hypoxic effect and activates HIF-1α, which then promotes glycolysis by upregulating the expression of several glycolytic enzymes, including hexokinase 2 [156,157], phosphofructokinase [158], phosphoglucomutase 1 [159], enolase [160], pyruvate kinase, pyruvate dehydrogenase (PDH), pyruvate dehydrogenase kinase (PDK) [161], lactate dehydrogenase A (LDHA) [160], monocarboxylate transporter 4 [162,163,164], and glucose transporters GLUT1 and GLUT3 [165]. Additionally, HIF-1α reduces the OXPHOS capacity by inhibiting mitochondrial biogenesis [166,167], decreasing PDH activity [161], and reducing ETC activity [168].

#### 2.1.4. Reason 4: Dysregulation of the Glycolytic Machinery

Studies have indicated that dysregulation occurs at the level of glycolytic protein expression (Table 3). Subsequent research has elucidated the role of the pyruvate dehydrogenase complex (PDC) in the metabolic switch in tumor cells towards aerobic glycolysis. In 2007, Koukourakis et al. observed a significant decrease in or absence of PDC expression and/or an overexpression of PDK in 91% of lung cancer patients tested via immunohistochemistry [8]. Typically, active PDC facilitates the oxidative decarboxylation of pyruvate into acetyl-CoA within the mitochondria. Conversely, PDK phosphorylates and deactivates PDC [16]. When PDC is inactive, pyruvate accumulates in the mitochondria and translocates back to the cytosol, where it is converted to lactate and NADH [16].

In this milieu, NADH was found to play a role in the glycolytic shift by directly or indirectly inhibiting PDC and activating PDK. Recent studies propose that a high concentration of cytosolic NADH, coupled with increased pyruvate, decreased lactate, and an active LDHA enzyme, positively promotes glycolysis in cancer [71,169,170,171].

Certain investigations have also demonstrated that overexpression of the antioxidant UCP2 in cancer cell lines promotes aerobic glycolysis, tumor proliferation, and resistance to apoptosis-induced chemotherapy [90,91,92] (see Figure 2). In 2016, Brandi et al. illustrated that UCP2 upregulates the expression of heterogeneous nuclear ribonucleoprotein A2/B1, which in turn regulates the transcription of GLUT1, pyruvate kinase M2 (PKM2), and LDH genes. UCP2 facilitates the metabolic shift in cancer cells towards Warburg’s aerobic glycolysis [92].

**Table 3 biomedicines-12-00211-t003:** Common alterations reported in glycolysis-related genes in cancer.

Gene ID	Gene Name	Mutation/Deregulation	Function in Glycolysis	Publication
*HK*	Hexokinase	Upregulated by p53 in cancer and promotes tumor growth and survival	Phosphorylates glucose when it enters the cells	[78,172,173,174]
*PFK1*	6-Phosphofructokinsae-1	Amplification and/or upregulation, posttranslational modification reported in multiple cancer types	PFK1 catalyzes the phosphorylation of fructose-6-phosphate (F6P) to fructose-1, 6-bisphosphate (Fru-1,6-P2) using Mg-ATP as a phosphoryl donor.	[78,175,176,177]
*PK*	Pyruvate kinase	Posttranslational modification or enhanced expression that benefits cancer	PK is involved in the final step of glycolysis, and it mediates the transfer of a phosphate group from phosphoenolpyruvate (PEP) to ADP, resulting in pyruvate and ATP.	[78,178,179,180,181]
*PDK-1*	Pyruvate dehydrogenase kinase-1	Upregulation	PDK is a kinase enzyme that inactivates pyruvate dehydrogenase by phosphorylation dephosphorylation at different specific serine residues. PDK decreases the oxidation of pyruvate in mitochondria and increases the conversion of pyruvate to lactate in the cytosol.	[78,182,183,184]

#### 2.1.5. Reason 5: AMPK Inhibition in Cancer Leads to a Glycolytic Shift

AMPK is a highly conserved serine/threonine protein complex that acts as a metabolic sensor and a master regulator of cellular metabolic homeostasis [185]. AMPK can be activated by either one of the two cell signals; the first is intracellular Ca^2+^-dependent, whereas the second is AMP-dependent (see Figure 4). AMPK is modulated either by phosphorylation or by allosteric activation. In response to an increase in the AMP/ATP ratio, liver kinase B1 (LKB1) is activated, which in turn directly activates AMPK by phosphorylation at Thr172, located in the catalytic subunit of the AMPK protein [186,187,188,189]. The AMP/ATP ratio can be altered during various intracellular states, such as hypoxia, glucose deprivation, calcium concentration, cytokines, and adipokines, and by certain hormones [189]. On the one hand, active AMPK inhibits the biosynthetic pathways in the cell, such as hepatic fatty acid synthesis and protein synthesis. On the other hand, active AMPK activates ATP-generating catabolic pathways, such as fatty acid uptake and oxidation, glycolysis, and mitochondrial biogenesis (see Figure 5). In cancer, AMPK is generally considered a tumor suppressor [190]. Studies have found that the dysregulation of AMPK plays a role in the glycolytic metabolic switch in cancer. Low AMPK expression is further implicated in tumorigenesis by promoting tumor initiation and progression [191]. Inactivation or reduced expression of AMPK in cancer promotes tumor growth and invasiveness [192,193]. This is an expected scenario, considering the activities exerted by AMPK.

A closer look at how AMPK alters the metabolic phenotype in cancer reveals that AMPK modulates mitochondrial respiration by activating autophagy (including mitophagy). This activation occurs through the phosphorylation and activation of ULK1, thereby regulating the localization of a crucial component of the phagophore known as autophagy-related protein 9 (ATG9) [194,195] (see Figure 5). mTOR, on the one hand, can inhibit ULK1/2, thus blocking autophagy [196]. AMPK can also induce mitochondrial biogenesis, aiming to increase the capacity of OXPHOS. Moreover, Faubert et al. (2013) found that AMPK negatively regulates the Warburg effect and suppresses tumor growth in vivo [197]. Faubert et al. documented that knocking down the α catalytic subunit of AMPK accelerates Myc-induced tumorigenesis. Furthermore, the inactivation of AMPK causes the stabilization of HIF-1α and a glycolytic shift in tumor cells in vitro. Altogether, the inhibition of AMPK in cancer inhibits OXPHOS and activates the Warburg effect in tumor cells (see Figure 5).

More interestingly, in cancer, AMPK acts through the AMPK/tuberous sclerosis complex (TSC)/mTOR signaling axis to regulate the metabolic switch. Inoki and colleagues found that active AMPK phosphorylates and activates TSC2 [198]. Earlier, Inoki had established that both TSC1 and TSC2 inhibit the activity of mTOR by suppressing the phosphorylation of ribosomal protein S6 kinase B1 (S6K) and eukaryotic translation initiation factor 4E-binding protein 1 (4E-BP1), which are downstream targets of mTOR [198,199]. Inoki et al. further reported that TSC1/2 inhibits the phosphorylation of S6K and 4E-BP1 by targeting Ras homolog (Rheb)—the protein that activates the protein kinase activity of mTOR [200]. They showed in vitro that TSC2 acts as a GTPase-activating protein that blocks the activity of Rheb and regulates its level [200]. In cancer, mTOR levels are often elevated and have been observed to stimulate aerobic glycolysis via the induction of pyruvate kinase isoenzyme 2 (PKM2) and other glycolytic enzymes [201]. A recent study by Ling et al. unveiled groundbreaking findings, reporting that mTOR directly inhibits AMPK by phosphorylating AMPK α1 at S347 and α2 at S345 in mammals. This inhibition is associated with a decreased phosphorylation of the activation loop T172. Interestingly, a reduction in mTOR activity resulted in AMPK activation independently of the AMP/ATP ratio [202].

In summary, active AMPK could activate autophagy either directly through the activation of ULK2 or indirectly by activating TSC2, thereby further inhibiting mTOR.

## 3. The Metabolic Shift in T2D

The metabolic switch in the case of insulin resistance and T2D occurs during two stages. The first stage occurs during the pathogenesis of insulin resistance and T2D, whereas the second stage occurs when both insulin resistance and T2D have already manifested clinically. The first stage is characterized by hyperactive OXPHOS and is due to high glucose uptake. In the second stage, distinct metabolic patterns arise in a tissue-specific manner. Conflicting reports on OXPHOS have been described in human and in vitro studies, showing either normal functioning mitochondria and active OXPHOS or dysfunctional mitochondria. However, evidence suggests that if mitochondrial dysfunction does occur, it happens as a result of, rather than being the cause of, insulin resistance and T2D.

### 3.1. The Metabolic Shift during the Pathogenesis of Insulin Resistance and T2D

Under normal conditions of insulin sensitivity, glucose-stimulated insulin secretion regulates glucose uptake and increases the activity of OXPHOS mitochondrial respiration [45,203]. During the pathogenesis of insulin resistance and towards the emergence of T2D, there is a state of chronic hyperglycemia caused by excessive nutrient supply and/or physical inactivity. Chronic hyperglycemia stimulates the continuous release of insulin from pancreatic β cells [204]. Insulin secretion increases glycolysis and pyruvate production [203] and elevates chronic hyperactivity of mitochondrial OXPHOS in response to insulin signaling. This heightened state of OXPHOS results in an augmented state of oxidative stress. In addition to the release of ROS, prolonged hyperglycemia causes glucose toxicity [37,38,39,40,205,206]. Both glucose toxicity and ROS ultimately damage pancreatic β cells and impair their ability to sufficiently secrete insulin [39]. Although ROS production partly results from enhanced mitochondrial respiration under glucose stimulation, the expression of antioxidant genes is unusually low in β cells, leading to ROS accumulation in the cytoplasm owing to inefficient ROS elimination [207]. Therefore, this is followed by a decrease in insulin secretion and is accompanied by a decrease in the rate of glycolysis [208]. In addition to the impairment of insulin secretion, ROS are major players in developing insulin resistance by rendering cells insensitive to insulin, thus hindering the insulin signaling pathway in peripheral tissues such as SKM, adipose tissue, and the liver [43,209]. This is because excessive ROS generation activates protein kinase (PK) signaling pathways [210]. Insulin signaling is thus suppressed downstream of the insulin receptor (IR) at the level of IR substrate-1 (IRS-1) and PI3K, which together promote insulin resistance in peripheral tissues [211,212].

### 3.2. The Metabolic Shift in Established Insulin Resistance and T2D

As mentioned earlier, accumulating evidence reveals that mitochondrial dysfunction is a result, rather than the cause, of insulin resistance in T2D [213,214]. The previous hypothesis, proposing insulin resistance as a result of mitochondrial dysfunction, relies mainly on association studies rather than cause-and-effect investigations. Several association studies employing animal models documented mitochondrial dysfunction concurrently with insulin resistance. The reported mitochondrial dysfunction was primarily tissue-specific, predominantly in the SKM of animal models or patients with T2D or insulin resistance, contrasting with the observation of active or normal OXPHOS in the liver of patients with T2D or insulin resistance.

Here, we bring forth examples of such association studies. For instance, OXPHOS genes were found to be downregulated in the SKM of patients with T2D [44,215,216] and after high-fat diet consumption [217]. Additionally, both Kelley et al. (2002) [50] and Mogensen et al. (2007) showed impaired mitochondrial respiration in the SKM of patients with T2D compared to their obese nondiabetic counterparts [218]. Several research groups reported contradictory findings in T2D, whereby the presence of normally functioning mitochondria in SKM was demonstrated [51,52,53].

In contrast to the SKM results, Takamura et al. (2006) found that genes encoding OXPHOS proteins are upregulated during fasting hyperglycemia in the livers of patients with T2D [54]. In mouse models, hepatic mitochondria adapted to a high-fat diet, preventing hepatic steatosis through increased OXPHOS activity and ETC uncoupling [219]. Collectively, these data support the rationale that mitochondrial respiration is regulated by different tissue-specific mechanisms, partially explaining the non-uniform response to excessive nutrients, obesity, physical inactivity, and insulin resistance across different tissues, organs, and research contexts. Moreover, other data have demonstrated that mitochondrial inhibition using drugs enhances insulin sensitivity and benefits patients with T2D.

### 3.3. Mitochondrial Dysfunction Is a Result of Insulin Resistance

Recent studies have explored the cause-and-effect relationship between insulin resistance and mitochondrial dysfunction using pharmacological and transgenic animal model approaches. The results corroborate that mitochondrial dysfunction is a consequence of insulin resistance, not the reverse. The resultant mitochondrial dysfunction was identified as a protective mechanism against insulin resistance. This was shown in a study by Pospisilik et al. in 2007, who induced OXPHOS deficiency by knocking out the AIF gene in mouse models. OXPHOS deficiency reduced fat mass, increased insulin sensitivity, and enhanced glucose tolerance. This study established that inhibiting OXPHOS did not induce insulin resistance in mice; instead, it was protective against insulin resistance development in obese mice [220].

Nair et al. showed that mitochondrial dysfunction and altered expression of mitochondrial genes are not intrinsic defects in patients with T2D but rather secondary to abnormal glucose and insulin secretion levels. Diabetic and nondiabetic individuals exhibited similar mitochondrial content, and after low-dose insulin infusion, both groups showed similar ATP production. High-dose insulin revealed lower ATP production in diabetic patients, along with reduced expression of peroxisome proliferator-activated receptor G coactivator-1 (PGC-1), citrate synthase, and cytochrome c oxidase [216].

In another study, Roden et al. found that accumulated intramyocellular fatty acyl-CoA causes the downregulation of OXPHOS genes by decreasing the expression of PGC-1. However, they suggested that the vicious cycle of metabolic changes in T2D starts with the increased availability of free fatty acids (FFAs), lipid accumulation in myocytes, and impaired lipid oxidation, which may cause mitochondrial dysfunction in the future [221].

An interesting study by Fazakerley et al. found that induced mitochondrial oxidative stress impaired glucose uptake, which was induced by insulin, and decreased the translocation of the GLUT4 protein to the cell membrane in adipocytes and myotubes of C57BL/6J mice. However, the induced mitochondrial oxidative stress did not alter the activity of OXPHOS [222]. Interestingly, a review by Lewis et al. (2019) on various experimental designs, which attempted to measure or assess mitochondrial OXPHOS in SKM, refuted the misconception that mitochondrial OXPHOS is dysfunctional and downregulated in T2D and proved instead that it is due to the limited oxygen supply to these tissues. They found that there are limitations in the reviewed in vivo and in vitro studies on the human mitochondrial SKM function [223]. The authors suggest that the mitochondrial respiratory capacity is intact in T2D when using high-resolution respirometry on isolated mitochondria and that other mitochondrial respiratory inadequacies detected in some in vivo studies are more likely due to changes in mitochondrial fractional volume [223]. These changes could be due to a less active lifestyle or limited oxygen availability in the cytosolic environment [81].

Interestingly, several other studies found that the mitochondrial OXPHOS was intact in T2D [52,223,224].

### 3.4. The Role of ROS in Insulin Resistance and T2D

From a mechanistic standpoint, an elevation in ROS levels can instigate the activation of stress-sensitive serine/threonine kinase signaling pathways, including c-Jun N-terminal kinase (JNK) [225], nuclear factor kappa B (NF-κB) [226,227], p38MAPK [228], and others. These pathways subsequently phosphorylate multiple targets, with IRS proteins being among them. The heightened serine phosphorylation of IRS diminishes its capacity for tyrosine phosphorylation, potentially hastening the degradation of IRS-1. This provides a plausible explanation for the molecular underpinnings of oxidative stress-induced insulin resistance. Compelling data affirm the crucial role of JNK activation, NF-κB kinase, protein kinase C inhibition, and potentially other stress- and inflammation-activated kinases in the development of oxidative stress-induced insulin resistance. These findings suggest that they could serve as appealing pharmacological targets to enhance insulin sensitivity [212,229].

Al-Mulla and Bitar et al. made a series of interesting discoveries, wherein they explored the role of oxidative stress in insulin resistance and T2D, along with their mechanism of action both in vitro and in vivo. In 2015, they found that oxidative stress and PKA activation were associated with diabetes in Goto–Kakizaki diabetic rat models. They showed that oxidative stress and PKA induced insulin resistance by enhancing cAMP-responsive element modulator/inducible cAMP early repressor (CREM/ICER) expression, which reduced IRS-2 expression by inhibiting the transcriptional activity of the cAMP response element (CRE) [230]. In another study, they corroborated that T2D instigates a cascade of events that produce ROS (mainly O_2_) from NADPH oxidase, leading to the oxidation of BH4 and uncoupling of NOS, which ultimately leads to NO inactivation with subsequent peroxynitrite formation. Altogether, an imbalance in the redox state is caused by increased ROS bioavailability and reduced antioxidant capability, which translates into a heightened state of oxidative stress [231,232]. Moreover, the authors demonstrated that the high oxidative stress in T2D is partly attributable to the diminished intracellular stabilization of NRF2 in dermal fibroblasts that were isolated and cultured from Goto–Kakizaki rats. Low NRF2 stabilization caused a decrease in the antioxidant effect of NRF2 in response to glucose-induced oxidative stress in dermal fibroblasts compared to cells in normoglycemic conditions [232]. Therefore, reduced NRF2 is also associated with higher cellular sensitivity to oxygen free radicals and results in cellular necrosis [232].

In 2012, Bitar and Al-Mulla found that ROS is responsible for the development of insulin-like growth factor 1 (IGF-1) resistance and, consequently, delayed wound healing in a T2D rat model [209]. IGF-1 resistance is another mechanism involved in developing insulin resistance in peripheral tissues. IGF-1 signaling, via the IGF-1 receptor (IGF-1R), uses downstream mediators that are commonly involved in the insulin signaling pathway. Altered IGF-1 function has been implicated in the pathogenesis of insulin resistance and in several other diseases, such as autoimmune diseases, atherothrombosis, osteoporosis, and certain common types of cancer [209]. In their study, IGF-1 activation in the PI3K-AKT-GSK-3ß pathway was attenuated in fibroblasts in vitro that had phenotypic features of diabetes or hypercortisolemia. In contrast, the ROS-activated JNK pathway led to the inhibitory phosphorylation of IRS1 at Ser307. Bitar and Al-Mulla showed that ROS, via the activation of JNK-p-IRS1 (Ser307), mediates IGF-1 resistance in T2D [209]. In 2019, Akhter et al. shed light on another mechanism by which oxidative stress is involved in inducing metabolic inflammation in T2D, i.e., through the upregulation of toll-like receptors (2 and 4), interferon regulatory factors (3 and 5), and other key pro-inflammatory cytokines in peripheral blood mononuclear cells. This mechanism depends on MAPK/NF-κß signaling [233].

### 3.5. AMPK Inhibition Is Implicated in Insulin Resistance and T2D

AMPK is regarded as the guardian of mitochondria, a complex master regulator and a key metabolic sensor. Paradoxically, AMPK is the link that connects the metabolic disturbances in cancer and diabetes, like the concept of Yin and Yang. Although AMPK is inhibited in both diseases, it exerts multi-faceted functions through different signaling pathways. Ultimately, low AMPK activity promotes tumor growth and proliferation and causes insulin resistance in the peripheral tissues of patients with T2D.

As previously discussed, AMPK controls mitochondrial biogenesis, dynamics, and disposal by mitophagy. Therefore, in low cellular ATP states, active AMPK restores ATP homeostasis by increasing mitochondrial ATP production, whereas low AMPK inhibits autophagy [234]. It was found that T2D is associated with suppressed autophagy and lipid accumulation [235].

AMPK plays an important role in the metabolic shifts associated with insulin resistance and T2D [185]. In animal studies, low AMPK activity contributed to the development of insulin resistance [236,237,238]. Inhibition of AMPK reduced glucose uptake and utilization due to a decline in the phosphorylation of target proteins involved in the trafficking of glucose transporters GLUT1 and GLUT4. These target proteins are thioredoxin interacting protein, TBC1 domain family member 1 protein, and phospholipase D1. Low AMPK also inhibits FFA β-oxidation in the mitochondria and exacerbates lipid biosynthesis, leading to the accumulation of lipids in cells and tissues. Normally, an active AMPK would enhance the breakdown of lipids by stimulating lipases. The activity of carnitine palmitoyltransferase I (CPT1), which transports FFA into mitochondria, is indirectly stimulated by AMPK. AMPK phosphorylates acetyl-CoA carboxylases 1 and 2, which in turn blocks the production of malonyl-CoA [238]. Malonyl-CoA is a potent inhibitor of the lipid transporter CPT1 [185,238]. Additionally, it was found that active AMPK inhibits hepatic gluconeogenesis by enhancing the expression of the orphan nuclear receptor small heterodimer partner (SHP) gene, which inhibits the transcriptional activity of cAMP-responsive element binding protein 1 (CREB). CREB regulates the transcription of hepatic gluconeogenesis genes [239].

Taken together, the inhibition of AMPK causes a metabolic shift in T2D through several mechanisms that decrease glucose utilization, inhibit FFA β oxidation, cause lipid accumulation in tissues, and activate hepatic gluconeogenesis. Generally, these changes are known to be implicated in developing insulin resistance (see Figure 6).

## 4. Metabolic Therapeutic Approaches in Cancer and T2D

Cancer and T2D are metabolic disorders characterized by opposing metabolic switches and divergent underlying signaling pathways, yet they intertwine towards the master regulator AMPK. In cancer, the Warburg glycolytic shift promotes malignant transformation, tumor progression, invasiveness, and resistance to chemotherapy and/or radiotherapy [121,240,241]. Nonetheless, during the pathogenesis of T2D, there is hyperactivity and dominance of mitochondrial OXPHOS. Therapeutic and/or nutritional targeting of either of the two metabolic shifts is a promising approach to correcting the metabolic imbalance and restoring homeostasis [21,241,242,243].

Although tumors are predominately glycolytic, they vary in their phenotypic features associated with proliferation, invasion, metastasis, and resistance to therapy. The characteristics of the metabolic phenotype for each cancer determine its rate of proliferation and resistance to chemotherapy [153]. Thus, cancer therapy needs to be customized to target the underlying causative metabolic dysfunction. Therapeutic attempts to target cellular metabolism in cancer are aimed at the inhibition of Warburg glycolysis and/or the activation of OXPHOS to confer antiproliferative activity. In addition, metabolic inhibition has shown the ability to sensitize chemo-resistant tumor cells to treatment. Furthermore, based on previous research, there have been suggestions to reestablish the metabolic imbalance in cancer by targeting tumor microenvironment symbiotic crosstalk.

Certain pharmacological agents and nutrients have been shown to have the potential to correct and reverse metabolic imbalances in cancer. Some of these are gaining validation through in vitro and in vivo analyses, as well as in clinical trials.

However, in T2D, targeted metabolic inhibition using nutritional and/or pharmacological compounds could prevent insulin resistance and improve insulin sensitivity in prediabetic and diabetic animal models and in diabetic patients. These therapies aim to inhibit mitochondrial OXPHOS activity [43,220]. The use of certain nutrients and dietary supplements as metabolic treatments or adjuvants in T2D is gaining attention owing to the encouraging results obtained in the past decade. In the next subsections, we summarize the pharmacological-based approaches to targeting mitochondrial metabolism in cancer and T2D.

In the upcoming sections, our focus is on pharmaceutical and nutritional approaches targeting metabolic imbalances in both cancer and T2D. Numerous other possibilities exist; the given examples are merely illustrative, showcasing the potential to counteract the metabolic shift and restore equilibrium.

### 4.1. Pharmacological-Based Approaches Targeting Mitochondrial Metabolism in Cancer

#### 4.1.1. BACH1 Depletion Activates OXPHOS and Sensitizes Tumor Cells to Metformin

Among the genes related to ROS homeostasis, BTB domain and CNC homolog 1 (BACH1) is a heme-binding transcription factor that combats the oxidative stress response by repressing the heme oxygenase 1 gene and is a negative regulator of ROS-induced cellular senescence directed by p53 [244,245]. BACH1 is upregulated in breast and other types of cancer; it is proposed to be a marker of poor prognosis and a high metastatic rate in breast cancer. For instance, triple-negative breast cancer (TNBC) cells reprogram their metabolism by increasing BACH1 expression to direct their metabolism away from the TCA cycle, which could be a protective mechanism that enhances their proliferative potential. On the one hand, it prevents the accumulation of ROS by shutting down mitochondrial metabolism. Thus, BACH1 may provide a mechanism by which tumor cells evade oxidative stress-induced senescence.

In 2019, Rosner et al. [246] showed that the combined therapeutic use of metformin with BACH1 inhibitor (hemin) could reverse chemoresistance in TNBC cells. BACH1 targets mitochondrial metabolism by repressing key ETC genes (UQCRC1 and ATP5D, both negatively correlated with BACH1 in TNBC), which are predominantly involved in the OXPHOS pathway. Metformin is known to mainly inhibit mitochondrial ETC complex I, along with other metabolic targets. Metformin was able to inhibit the growth of tumor cells and decrease tumor cell viability in BACH1-depleted TNBC cells. However, control cells that expressed BACH1 did not respond to metformin treatment, and the TNBC cells continued to grow and proliferate. Downregulating BACH1 in tumors using hemin, both in vitro and in vivo, resulted in an increased expression of mitochondrial inner membrane genes involved in ETC and promoted mitochondrial respiration. TNBC cells that were depleted of BACH1 exhibited higher oxygen consumption, lower lactate production, higher glucose utilization in the TCA cycle, increased ATP generation, higher TCA cycle intermediate production, and decreased glycolysis-related intermediates [246]. Rosner attempted to reprogram the metabolic pathway in TNBC tumors resistant to ETC inhibition therapy because of high BACH1 expression. Inhibiting BACH1 expression sensitized tumor cells to metformin both in vitro and in vivo. For further details regarding interventional clinical trials investigating the impact of metformin on various types of cancer, we compiled a table retrieved from clinicaltrials.gov on 23 November 2023. This table encompasses both completed and ongoing studies that have reached phase 2 or phase 3. It is important to note that trials that were withdrawn, suspended, or terminated were excluded (refer to Appendix A).

Cellular senescence is mainly mediated by tumor suppressor p53, which serves as a barrier to the malignant transformation [247]. The upregulation of BACH1 in TNBC cells has been suggested to prevent oxidative stress-induced senescence. This rationale is supported by the findings of Dohi et al., who demonstrated that BACH1 forms a complex with p53, histone deacetylase 1, and nuclear co-repressor. The formation of this complex prevents p53 from inducing an effective oxidative stress response by promoting histone deacetylation [245]. Furthermore, Wiel et al. showed that stabilizing BACH1 using antioxidants in a p53-/- background in lung cancer models increased metastasis, glucose uptake, glycolysis rate, and lactate secretion in mouse and human lung cancer cells. Hence, in scenarios marked by lower oxidative stress, BACH1 promotes glycolysis-dependent lung cancer metastasis independently of p53 [248]. Multiple microRNAs (miRs) were found to target the post-transcriptional regulation of BACH1 and reduce cancer progression, such as miR-142-3p, which can target BACH1 in breast cancer cells, leading to reduced cellular proliferation, invasion, and migration [249]. The induction of miR-330 also inhibits the proliferation of colorectal cancer cells by suppressing BACH1 gene expression [250].

In addition to these studies, BACH1 was also found to be linked to an age-dependent decline in adaptive homeostasis. Its levels were elevated in various tissues, including the heart, liver, and lungs, in aging mice [247]. Furthermore, BACH1 expression was higher in human bronchial epithelial cells obtained from older adults compared to those from young adult donors [251]. Thus, BACH1 attenuates adaptive redox homeostasis in both aging mice and older individuals. Taken together, these studies show that BACH1 is a potential metabolism-targeting therapy for cancer. This suggests that the inhibition of BACH1 can modulate the metabolic profile in resistant cancers such that the OXPHOS pathway is restored, glycolysis is reduced or omitted, cancer growth is halted, and cancer cells are sensitized to therapy.

#### 4.1.2. Dichloroacetate and EGFR-Inhibitors Reverse the Warburg Effect in Cancer

Sun et al. demonstrated that the generic drug dichloroacetate (DCA) can reverse the glycolytic phenotype in metastatic breast cancer cells both in vitro and in vivo and can inhibit tumor growth and metastasis [252]. DCA works by inhibiting PDK activity, wherein PDK inactivates PDH via phosphorylation. PDH controls the conversion of pyruvate to acetyl Co-A, which in turn enters the TCA cycle and generates ATP via the action of OXPHOS. Thus, treatment with DCA stops the inhibition of PDH, increases the flux of pyruvate into the mitochondria, and promotes mitochondrial OXPHOS over glycolysis [252].

In 2015, De Rosa et al. demonstrated that the use of EGFR inhibitors, including erlotinib or WZ4002 in human non-small cell lung cancer cell lines (H1975, HCC827, and H1993) and PHA-665,752 in the H1993 cell line, succeeded in the reversal of the Warburg effect and reactivation of OXPHOS in these cell lines [253]. This effect was mediated through the upregulation of ETC mitochondrial complexes, in addition to reduced expression levels of key glycolysis enzymes, such as hexokinase II and p-PKM2 Tyr105. Concomitantly, decreased lactate secretion and increased intracellular ATP levels were observed in response to EGFR inhibition [253]. In conclusion, these results revealed that the effective inhibition of EGFR signaling can reverse the Warburg effect in cancer cell lines and restore OXPHOS.

#### 4.1.3. Metformin Activates AMPK to Induce Apoptosis in Cancer

Targeting AMPK in cancer cells to either sensitize tumor cells to chemotherapy, cause cell cycle arrest, or induce apoptosis are promising therapeutic approaches. For instance, the activation of AMPK inhibited cervical cancer cell proliferation through AKT/FOXO3a/FOXM1 signaling cascade by counteracting the function of Forkhead box M1 (FOXM1) [192]. Previously, several pharmacological AMPK activators, such as metformin, the AMP-mimetic 5-aminoimidazole-4-carboxamide (AICAR), and the ATPase inhibitor A23187, were able to suppress cervical cancer cell growth by activating AMPK [192].

In 2008, Keith et al. were able to induce cell cycle arrest in the MDA-MB-231 breast cancer cell line by treating the cells with metformin only in the presence of cyclin-dependent kinase inhibitors (p27^kip^ and/or p21^cip1^). Metformin was able to activate the AMPK pathway and downregulate cyclin D1 [254]. Mills et al. further demonstrated that the LKB1-AMPK pathway regulates p27^kip1^ phosphorylation; they were able to induce apoptosis in cell lines after AMPK activation in the absence of p27^kip1^. Downstream of AMPK, p27^Kip1^ is phosphorylated at Thr198, which stabilizes p27, leading to autophagy and cell-cycle progression. When p27 was knocked down in the cancer cell line, LKB1-AMPK activation induced apoptosis [255].

#### 4.1.4. Targeting PI3K/AKT Pathway in Cancer

The dysregulation of the PI3K/AKT pathway is a common feature in many cancers [256]. Evidence indicates that inhibiting the PI3K/AKT pathway hinders tumor progression [256]. However, the use of PI3K-AKT-mTOR inhibitors in treating various cancer types has been observed to induce hyperglycemia in patients [257]. A study by Khan et al. investigated the clinical data of 341 cancer patients from 12 phase I clinical trials treated with PI3K, AKT, or mTOR inhibitors as well as dual inhibitors. There was evident hyperglycemia in 87.4% of these patients. However, grade-three hyperglycemia was only seen in 6.7% of these patients. Hence, hyperglycemia was mostly manageable in those patients. Thus, caution is necessary when treating cancer patients who are also diabetics with PI3K-AKT-MTOR inhibitors [257]. This study may seem paradoxical, as the inhibition of the PI3K-AKT-mTOR pathway, which generally leads to inhibition of cell proliferation, is expected to activate the AMPK pathway. AMPK activation would exhibit beneficial effects in diabetes and lower glucose levels. However, this is not the case with PI3K-AKT-mTOR inhibitors alone. Nevertheless, the combination of metformin and PI3K-AKT-mTOR inhibitors in vitro enhances apoptosis of ovarian cancer cells [258] and induces drug sensitivity in pancreatic cancer cells [259].

To understand how PI3K-AKT-mTOR inhibitors work, we will take a quick look at PI3K signaling. PI3K produces phosphatidylinositol (3,4,5)-trisphosphate (PIP3), which in turn activates phospholipase D (PLD) [260]. PLD catalyzes the hydrolysis of the membrane phospholipid phosphatidylcholine to generate choline and metabolically active phosphatidic acid (PA) [261]. PA is a signaling lipid involved in processes such as cell proliferation and vesicular trafficking. PLD can influence mTOR activity by generating PA [262], which directly activates mTOR complex 1 (mTORC1) under certain conditions [263]. PA stimulates mTORC1 function and suppresses the activation of mTORC2 as part of a mTORC1/2 feedback loop [264]. PI3K inhibitors decrease PLD activation after insulin receptor stimulation [265], and the mutation of the PIP3 binding site on PLD prevents PLD activation and membrane recruitment [266]. A study by Toschi et al. demonstrated that by inhibiting PLD activity, mTORC2 could be targeted therapeutically with rapamycin [267]. Thus, the combination of rapamycin, metformin, and PI3K/PLD inhibitors can have a favorable therapeutic outcome in cancer therapy.

PIP3 generation mediates downstream signaling events that inhibit glycogen synthase kinase-3β (GSK-3β) [268,269]. GSK-3β in turn hinders NRF2 by directing it towards ubiquitination and subsequent degradation [270]. NRF2 plays a pivotal role in combating oxidative stress and regulating redox homeostasis, thereby safeguarding cells against carcinogenesis [271]. However, studies over the last decade reveal a “dark side” of NRF2 [272], where its constitutive stabilization leads to increased glutaminolysis [273], cancer progression [274], metastasis [275], and chemoresistance [276,277]. Indeed, NRF2 redirects glucose and glutamine into anabolic pathways during metabolic reprogramming [273,278]. Consequently, strategies such as inhibiting PI3K and NRF2 or activating GSK-3β, along with NRF2 repressor Kelch-like ECH-associated protein 1 (KEAP1) [279], hold promising therapeutic potential against cancer.

### 4.2. Pharmacological-Based Approaches Targeting Mitochondrial Metabolism in T2D

#### 4.2.1. Apoptosis-Inducing Factor Ablation in Diabetic Mice Inhibited OXPHOS

A study by Penninger‘s team in 2007 showed that global or tissue-specific gene ablation (liver and muscle) of apoptosis-inducing factor (AIF) in mice caused a deficiency in OXPHOS, which was accompanied by improved glucose tolerance, increased insulin sensitivity, and reduced fat mass [220]. AIF has been known to cause progressive OXPHOS dysfunction in mice [280,281]. Mutation analysis performed in several model organisms found that AIF was an essential regulatory gene for maintaining fully active and functional mitochondrial ETC [282,283]. Therefore, AIF deletion caused a progressive loss of ETC activity and function [280,282]. In the study by Penninger’s team, impaired OXPHOS prevented weight gain, insulin resistance, and T2D, which is contrary to other studies reporting that OXPHOS deficiency is associated with insulin resistance and T2D [220].

#### 4.2.2. Targeting PI3K/AKT Pathway in T2D

Su et al., in a comprehensive review, explain the effects of PI3K-AKT signaling on obesity and T2D. The review summarizes the findings of many studies done in vitro and in vivo on diabetic cells and mouse models in which the activity of the PI3K-AKT was targeted [284].

Su et al. argue that, under normal physiologic conditions, the PI3K-AKT pathway actively regulates body functions, including metabolism and proliferation. The PI3K-AKT pathway regulates glucose metabolism through FOXO1 and GSK-3. PI3K-AKT also regulates lipid metabolism through mTORC1 and SREBP. Active AKT inhibits FOXO1, which reduces glucose levels [285,286]. Similarly, active AKT inhibits mTOR complex 1, which consequently reduces lipid and protein production [287]. GSK-3 is also inhibited by AKT, which leads to glycogen synthesis, thus reducing glucose levels [288]. Lipid metabolism is regulated by AKT activity through sterol regulatory element-binding proteins (SREBP). SREBP increases fatty acid and cholesterol accumulation [284,287,289].

However, when there is chronic excessive energy intake, as in obesity, PI3K-AKT signaling becomes suppressed, a state in which re-activating PI3K-AKT would lessen obesity and insulin resistance. Nevertheless, it is in the established disease states of cancer and/or obesity where there is dysregulation and/or overexpression of PI3K-AKT. At this point, therapeutic inhibition of PI3K-AKT becomes an effective anti-obesity and anti-cancer treatment approach [284].

Thus, Su et al. detail the mechanism by which the PI3K-AKT pathway acts in an organ-specific manner [284]. That further explains why targeting PI3K, whether by inhibition or activation, would be favorable depending on the context [284]. For instance, one study showed that pharmacological inhibition of PI3K-AKT activity reduced adiposity and metabolic syndrome in obese mice and rhesus monkeys [290]. They used two small molecules with selective inhibitory action on PI3K (CNIO-PI3Ki and GDC-0941) as pharmacological inhibitors [290]. In contrast, overexpression of FAM3A in the liver activates PI3K p110α-AKT signaling in the liver and decreases hepatic gluconeogenesis and lipogenesis [291].

#### 4.2.3. Metformin as a Metabolic Inhibitor in T2D

For decades, metformin has shown great success in the treatment of T2D. Metformin can stimulate glucose uptake and glycolysis in patients with T2D [203]. Glycolysis plays two major roles in glucose homeostasis. The first role is through inhibiting hepatic gluconeogenesis, thereby decreasing the amount of glucose released into the blood [203,292], and the second role is through enhancing insulin secretion by pancreatic β cells [203,208,293]. In this context, metformin works by augmenting glycolysis, which leads to a decrease in liver gluconeogenesis. Metformin exerts its effects by suppressing mitochondrial OXPHOS by inhibiting complex I (NADH dehydrogenase) of the ETC [294,295]. Inhibiting complex I increases the AMP/ATP ratio, which further activates AMPK [185]. The idea behind metabolic inhibition is that any injury caused to the mitochondrial metabolic machinery leads to the activation of AMPK to compensate for the mitochondrial dysfunction. Metformin was also found to exert its function by upregulating UCP2 in adipocytes in mouse models, thus playing a protective role against oxidative damage [296]. Therefore, AMPK activation is an effective therapeutic strategy for enhancing insulin sensitivity in T2D [43].

### 4.3. Metformin and Other AMPK-Activators in Cancer Clinical Trials

Metformin, a standard anti-diabetic medication, has been an attractive therapeutic target in cancer patients. Clinical data on the effect of metformin and other AMPK activators in cancer patients strengthen our argument about targeting the metabolic shifts in both diabetes and cancer. Several meta-analysis studies over the last decade have reported that diabetic patients receiving metformin are at lower risk of developing cancer [3,297,298]. Moreover, metformin was able to improve survival and response to treatment in cancer patients [297,298,299,300]. These studies corroborated previous in vitro and in vivo studies in animal models that showed metformin exhibiting anti-cancer effects [192,297,298,301]. For instance, Noto et al. (2012) conducted a systematic meta-analysis on 6 studies (4 cohort studies, 2 RCTs), with data from a total of 210,829 diabetic patients [3]. They found that diabetic patients taking metformin had a significantly lower risk of cancer incidence and cancer mortality using pooled relative risk measures. One of the earliest meta-analysis studies, conducted by DeCensi et al. in 2010, showed a 31% reduction in overall relative risk of cancer incidence in subjects receiving metformin compared to other anti-diabetic treatments [302]. Another meta-analysis study, by Wang et al. (2014) [303], performed on data from 13 observational studies (10 cohort, 3 case-control), found that the use of metformin was associated with reduced risk of pancreatic cancer in T2D patients. In another observational study, by Kim et al. (2020), involving a Korean cohort of 323,430 individuals with a median follow-up of 12.7 years, data were extracted from national health records spanning from 2002 to 2015. The findings indicated that diabetic individuals undergoing metformin treatment had a reduced risk of cancer incidence compared to diabetic patients not receiving metformin, with an incidence percentage of 10.3% in metformin users compared to 11.1% in non-metformin users [304]. Similar results have been reported in other retrospective meta-analysis studies [305,306,307,308,309,310]. However, a study conducted in the UK showed no protective effect of metformin against cancer incidence in diabetic patients [311]. An insightful review by Saraei et al. (2019) aimed to explain the mechanisms by which metformin exerts its beneficial effects in cancer [297]. The review also encompassed clinical trials conducted to confirm the beneficial effects of metformin on cancer. Based on this analysis, several clinical trials took place in non-diabetic patients to test the effects of metformin, but the results were inconclusive in proving a protective anti-cancer effect in non-diabetic patients. Therefore, further investigations are needed.

An inquiry arises regarding the potential anti-cancer effects in diabetic patients of other AMPK activators similar to metformin. Although metformin is extensively studied as an AMPK activator, there exist additional physiological and pharmacological agents that can activate AMPK either directly or indirectly. For instance, thiazolidinediones (TZDs), such as troglitazone, pioglitazone, and rosiglitazone, belong to another class of anti-diabetic medications recognized for their ability to activate AMPK.

Some studies showed the absence of any significant association between cancer risk and taking TZDs in diabetic patients [312,313,314]. Other studies have shown that T2D patients who are taking TZD have lower cancer risk in certain cancer types [312,314,315,316]. Interestingly, some clinical studies have shown that patients taking TZD have an increased risk of cancer [317,318]. Thus, observations and associations have been conflicting and inconclusive in meta-analysis studies. This is attributed to methodological variations within these studies and the intricacy of the disease [319]. Thus, more studies are needed. However, the evidence that associates metformin use in T2D patients with a lower risk of cancer is stronger and more consistent among studies [319]. Although metformin (belonging to biguanides) and TZDs can indirectly activate AMPK by inhibiting complex 1 in the mitochondrial respiratory chain (ETC cycle), metformin also acts in a non-AMPK-dependent manner [320]. Metformin’s impact on the liver is mediated by antagonizing glucagon signaling through cyclic AMP and PKA, operating independently of AMPK [321,322]. In contrast, TZDs activate AMPK by targeting the nuclear hormone receptor peroxisome proliferator-activated receptors (PPARs), which in turn stimulate the secretion of adiponectin and, consequently, activate AMPK [323]. Other AMPK activators, such as polyphenols and 5-aminoimidazole-4-carboxamide riboside (AICAR), have not been studied in the context of anti-diabetic drugs and the risk of cancer. Yet, it is worth noting that similar anti-cancer effects to those of metformin have also been observed with other AMPK activators in vitro and in vivo. This suggests that metformin may not be the only drug with dual effects and that other AMPK activators might exhibit promising anti-cancer effects as well as anti-diabetic ones [323].

## 5. Nutritional Therapeutic Approaches in Cancer and T2D

In this context, we highlight three dietary compounds suitable for oral consumption: alpha-lipoic acid (ALA), flavonoids, and glutamine. Acknowledged for their minimal to no side effects, these compounds have demonstrated promising results in both in vitro and in vivo studies, as well as in combination therapies for individuals with diabetes and cancer [324,325,326,327]. Nevertheless, these are just a subset of various alternative nutritional approaches, including omega-3 polyunsaturated fatty acids (ω-3 PUFA) and artemisinin, which are beyond the scope of this review.

### 5.1. Nutritional- and Dietary-Based Approaches Targeting Mitochondrial Metabolism in Cancer

#### 5.1.1. Alpha-Lipoic Acid as a Metabolic Modulator in Cancer

ALA is a naturally occurring dithiol compound that is produced physiologically in the body from octanoic acid in the mitochondria. It can also be found in a variety of food and dietary supplements. ALA, through its various metabolic regulatory effects, can inhibit the proliferation, migration, and invasion of tumor cells and can induce apoptosis [328]. ALA has been well known for acting as a metal chelator and an ROS scavenger [328,329,330]. More importantly, ALA acts as a cofactor for several enzyme complexes, such as PDC, and promotes mitochondrial respiration [328]. Earlier studies reported that treatment with ALA decreased serum levels of pyruvate and lactate in both lean and obese individuals with T2D [331]. Later, ALA was reported to increase PDC activity in rat hepatocytes and in the mitochondria of hepatocytes in diabetic rat models [332,333]. In 2004, Patel et al. reported that ALA could minimize or block the inhibitory phosphorylation of the E1 subunit of the PDC complex via pyruvate kinase, thereby increasing the activity of PDC [332].

ALA is well known for its antioxidant action, which increases glutathione peroxidase activity and in turn reduces oxidative stress [329,334,335]. These antioxidant effects were seen in advanced-stage cancer patients who were administered ALA treatment for 10 consecutive days [335,336]. Nevertheless, ALA also plays the role of a prooxidant by increasing the production of free oxygen radicals in the mitochondria of colon cancer cell lines but not in non-transformed cells [334]. This prooxidant effect is a result of ALA stimulating mitochondrial OXPHOS and inducing a cytotoxic effect on cancer cells in both in vitro and in vivo models [328,337]. In 2005, Wenzel et al. discovered that ALA induced mitochondrial OXPHOS in a colon cancer cell line (HT-29) and stimulated apoptosis [334]. ALA-induced apoptosis occurred selectively in colon cancer cells but not in the non-transformed cells [334]. Moreover, ALA-induced apoptosis ensued predominantly via the intrinsic mitochondrial apoptotic pathway and was independent of p53 [338]. Wenzel et al. further reported that ALA, along with its reduced form, i.e., dihydrolipoic acid (DHLA), was able to trigger apoptosis in cancer cells by increasing the production of mitochondrial ROS following an increased influx of lactate and pyruvate into the mitochondria. This effect was also associated with the downregulation of antiapoptotic protein BCL-X_L_ [334]. While studying two ovarian cancer cell lines (cisplatin-resistant and cisplatin-sensitive), Kafar et al. showed that ALA treatment induced apoptosis by downregulating the gene expression of antiapoptotic genes MCL-1 and BCL2L1 and by upregulating the expression of Bim, a pro-apoptotic gene [339]; another study showed similar results [329]. In line with these findings, Kim et al. reported that ALA treatment caused apoptosis in a dose-dependent manner in an in vitro setting in the MDA-MB-231 breast cancer cell line [329]. ALA promoted apoptosis by increasing the mRNA and protein expression of BAX and by decreasing the mRNA and protein expression of BCL2 [329]. In other cellular contexts, ALA prevented apoptosis [340,341]. Interestingly, ALA successfully reversed the Warburg effect and inhibited glycolysis through inhibition of PDK [325,342].

ALA has also been shown to modulate mitochondrial metabolism through the activation of AMPK signaling. Shen et al. (2007) revealed that ALA activated AMPK with an increased phosphorylation of AMPK at Thr172 in C2CL2 myotubes. Shen and colleagues found that ALA acted by enhancing Ca^+2^/calmodulin-dependent protein kinase kinase (CAMKK) and not through AMP-LKB1 signaling [343]. ALA activation of AMPK and the subsequent inhibition of mTOR-S6 signaling suppressed thyroid cancer cell proliferation in vivo in several thyroid cancer cell lines, including BCPAP, HTH-83, CAL-62, and FTC-133 [330]. Additionally, ALA decreased the migration and invasion of cancer cells in thyroid cancer cell lines by inhibiting transforming growth factor β (TGFβ) production and signaling cascade [330]. ALA further induced cell cycle arrest through the upregulation of cyclin-dependent kinase inhibitors p27^kip1^ and p21^cip1^ [344].

Altogether, studies point towards a pleiotropic effect of ALA on cancer cells depending on the type of cell and tumor.

#### 5.1.2. Flavonoids as a Metabolic Modulator in Cancer

Similar to ALA, flavonoids are a family of natural polyphenolic compounds found in fruits and vegetables that have promising anticancer effects [326,327,345]. They appear to modulate mitochondrial metabolism in cancer and reverse Warburg glycolysis [346,347,348]. In 2017, Wei et al. reported that didymin, a natural flavonoid, inhibited the proliferation of the liver cancer cell line HepG2 by decreasing cyclin B1, cyclin D1, and cyclin CDK4 [346]. Didymin also induced apoptosis in HepG2 cells by altering the BCL-2/BAX ratio and by stimulating caspase-mediated apoptosis. Moreover, Wei et al. showed that didymin could downregulate the ERK/MAPK and PI3K/Akt pathways by upregulating the Raf kinase inhibitory protein (RKIP). This study confirmed earlier observations made by Singhal et al. (2012), which demonstrated both in vivo and in vitro that didymin induced G2/M arrest and apoptosis in neuroblastoma cells and upregulated RKIP [349]. Zhao et al. recently reported that brosimone I, another flavonoid, induced apoptosis and cell cycle arrest through ROS-mediated endoplasmic reticulum stress and AMPK pathway activation in the human colon cancer cell line HCT116. The activation of AMPK depended on an increase in Ca^+2^ ions and the activation of the CaMKKβ-AMPK pathway but not on AMP [347]. Reportedly, other members of the flavonoid family have reversed Warburg glycolysis and promoted OXPHOS in several in vitro and in vivo preclinical cancer studies [350]. Table 4 summarizes a few of these findings of a group of flavonoids.

Although preclinical studies have positioned dietary flavonoids as potential candidates for treating and/or preventing cancer [345,350,374], these supplements have not yet shown substantial efficacy in clinical trials. In 2020, Bisol et al. published a systematic review of clinical trials where flavonoids were studied as potential therapeutic agents in cancer [375]. They identified 22 phase 2 clinical trials and 1 phase 3 clinical trial that administered flavonoids as either monotherapy or in combination with other chemotherapeutic agents. Twelve of these clinical trials enrolled patients with solid tumors, whereas the other eleven trials included patients with hematopoietic or lymphoid malignancies [375]. Overall, low rates of complete or partial response to flavonoid treatment were reported in clinical trials. Additionally, positive outcomes were mostly associated with hematopoietic or lymphoid tumors compared to solid tumors [375]. These clinical trials had various limitations, including small sample sizes and variations in the administered doses, design of the randomized trials, and tumor subtypes of the patients [375]. In addition to this, the limited bioavailability and varied absorption of administered flavonoids further limit its efficacy [376]. Studies are being performed to improve the bioavailability of flavonoids by enhancing the metabolic stability and absorption of the administered flavonoids [376,377].

Therefore, a greater number of well-designed clinical trials is required to test the efficacy of flavonoids for cancer treatment.

#### 5.1.3. Glutamine as a Nutritional Supplement in Cancer

Several studies have demonstrated the beneficial effects of glutamine in cancer both in animal models and cancer patients. For example, Martins et al. found that supplementation with 2% l-glutamine in Walker-256 tumor-bearing rats prevented tumor growth and cancer-associated cachexia while restoring cell proliferation in the normal intestinal mucosa [378].

Another study by Chang et al. suggests that glutamine supplementation in advanced non-small cell lung cancer (NSCLC) patients undergoing concurrent chemoradiotherapy prevented radiation-induced injury and weight loss [379]. Similarly, Pehlivan and colleagues showed that glutamine supplantation in NSCLC patients receiving concurrent chemoradiotherapy showed that glutamine reduced the incidence and severity of radiation-induced esophagitis, improved survival, and prevented weight loss. Interestingly, it did not negatively impact tumor growth [380]. Generally, several studies support the use of glutamine supplementation in combination with standard treatments to alleviate chemo- and radiotherapy-associated side effects, leading to improved outcomes [381]. Another approach suggested by Kodama et al. is to target glutaminase (GLS1) and phosphoribosyl pyrophosphate amidotransferase (PPAT) enzymes, rebalancing the PPAT/GLS1 enzyme ratio. Restoring GLS1 expression and/or downregulating PPAT enzyme might be effective in redirecting glutamine metabolism and inhibiting tumor growth [35].

### 5.2. Nutritional- and Dietary-Based Approaches Targeting Mitochondrial Metabolism in T2D

Accumulating evidence shows that ALA, flavonoids, and glutamine have beneficial effects as adjuvant and dietary supplements for the treatment of patients with T2D.

#### 5.2.1. ALA as a Metabolic Modulator in T2D

ALA functions as an antioxidant and an anti-inflammatory agent and is reportedly beneficial in treating patients with T2D. It exerts its antioxidant effects by quenching ROS, chelating metallic ions, and reducing the oxidized forms of glutathione, vitamin C, and vitamin E. Moreover, it boosts antioxidant machinery by enhancing NRF-2-mediated antioxidant gene expression. It also acts by activating AMPK in SKM and inhibiting NFκB [324]. Moreover, ALA activates hepatic AMPK, leading to decreased gluconeogenesis and glucose output from the liver [382]. It also activates AMPK in the SKM, which leads to an increase in glucose uptake and fatty acid oxidation [383]. Research suggests that ALA activates AMPK in the liver and SKM through increased intracellular calcium ion concentration and not through LKB1 activation [343]. Surprisingly, ALA was found to inhibit hypothalamic AMPK, leading to a reduction in food intake and body weight [324]. These metabolic effects of ALA have been tested in pre-diabetic volunteers in a randomized, placebo-controlled pilot study [384]. Twelve volunteers who were eligible and met the criteria for prediabetes were included in the study and took ALA supplementation (600 mg/day) for 30 days. ALA improved glycemic control and insulin sensitivity in pre-diabetic volunteers (assessed by HOMA-IR and fasting serum insulin); it, however, did not affect the lipid profile. In another study, diabetic individuals who orally consumed 600 mg of ALA supplementation (twice a day) also showed improved insulin sensitivity, assessed by a 2 h manual hyper-insulinemic euglycemic clamp technique, expressed as a glucose disposal rate and insulin sensitivity index [385]. Moreover, patients with diabetic nephropathy were shown to benefit from the oral administration of ALA [386].

#### 5.2.2. Flavonoids as a Metabolic Modulator in T2D

Flavonoids have demonstrated beneficial effects in T2D via metabolic reprogramming in pancreatic β-cells, hepatocytes, adipocytes, and SKM [326].

In an in vitro study by Kyriakis et al., two flavonoids, gallic acid and its dimer ellagic acid, were found to bind to glycogen phosphorylase and inhibit its action, thus decreasing glycogen metabolism and glucose production [387]. Therefore, it was suggested that gallic acid and ellagic acid could be administered as antihyperglycemic agents.

Lagouge et al. showed that mice fed a high-fat diet and administered a flavonoid called resveratrol did not develop obesity or insulin resistance. Resveratrol induced mitochondrial OXPHOS and improved muscle respiratory capacity by activating peroxisome proliferator-activated receptor gamma coactivator 1-alpha (PGC-1α) through sirtuin-1 (SIRT-1)-mediated deacetylation [388].

Recently, the therapeutic effects of flavonoids were further confirmed by Meng et al. [389]. Their study showed that flavonoids extracted from mulberry leaves activated AMPK in mice with spontaneous T2D and enhanced glucose uptake and OXPHOS in the L6 SKM cell line. Flavonoids also induced the expression of PGC-1α and the upregulation of GLUT4 [389].

Generally, flavonoids were reported to enhance insulin sensitivity, decrease ROS, and mitigate inflammation in SKM and adipose tissues [326]. Flavonoids could enhance insulin secretion by pancreatic β cells and reduce apoptosis in these cells. They also enhanced glucose uptake by SKM and white adipose tissue [326]. These encouraging results need to be further navigated and confirmed in clinical trials to be administered for therapeutic purposes in patients with T2D.

#### 5.2.3. Effects of Glutamine Supplementation on T2D

The glutamine pathway plays a protective role in T2D. Studies have shown that glutamine metabolism plays an important role in insulin signaling and glucose metabolism [390]. Specifically, the TCA cycle intermediates generated from glutamine metabolism can stimulate insulin secretion and enhance insulin sensitivity in various tissues, including the liver, muscle, and adipose tissue [391,392,393,394]. For example, α-ketoglutarate has been shown to stimulate insulin secretion in pancreatic beta cells. α-Ketoglutarate is converted to succinyl-CoA, which in turn activates the ATP-sensitive potassium channel, leading to depolarization of the cell membrane and subsequent calcium influx. The calcium influx triggers insulin secretion from the beta cells [395]. In addition, other TCA cycle intermediates, such as citrate and malate, have also been shown to stimulate insulin secretion. Citrate can enhance insulin secretion by activating the exocytotic machinery in pancreatic β cells [396], whereas malate can increase ATP production and stimulate insulin secretion [397].

Interestingly, α-ketoglutarate and succinate can activate the mTOR signaling pathway, which in turn enhances insulin signaling and glucose uptake [398,399]. The activation of mTOR stimulates the activity of IRS-1, a key signaling molecule in the insulin signaling pathway. This enhances the translocation of GLUT4 to the membrane and increases glucose uptake in insulin-sensitive tissues such as skeletal muscle and adipose tissue [400].

In addition to mTOR signaling, TCA cycle intermediates can also enhance insulin sensitivity by regulating the activity of key metabolic enzymes. For example, α-ketoglutarate can induce the activity of PDH, thus enhancing glucose oxidation and improving insulin sensitivity [401]. Likewise, succinate has been shown to inhibit the activity of HIF-1α. In addition to its role in cancer, it plays a role in glucose metabolism and insulin sensitivity [107,402]. Previous studies indicate a possible impact of glutamine on oxidative stress and inflammatory markers. In animal studies, supplementation with glutamine demonstrated a notable elevation in antioxidant proteins such as superoxide dismutase, glutathione peroxidase (GPx), and catalase levels [403,404,405,406], along with significant improvements in levels of inflammatory markers such as c-reactive protein, interleukins 6 and 23, and monocyte chemoattractant protein-1 [407]. The antioxidant effect of glutamine may be attributed to its involvement in glutathione synthesis, leading to increased enzymatic activity of GPx and a reduction in ROS production [403,404].

A comprehensive systematic review conducted by Maleki’s team in 2020 revealed interesting findings. Among the 19 examined studies, nine highlighted a significant increase GLP-1 levels in the sera. Furthermore, eight studies showed a reduction in fasting blood sugar levels, with four studies reporting decreases in postprandial blood sugar and triglyceride levels after glutamine supplementation. Although seven studies demonstrated a significant increase in insulinemia with glutamine, the outcomes regarding Hb-A1c levels were inconclusive [408].

Overall, the TCA cycle intermediates generated from glutamine metabolism can enhance insulin sensitivity by regulating multiple signaling pathways and metabolic enzymes. Understanding the complex interplay between glutamine metabolism and insulin signaling may provide insights into the development of new therapies for insulin resistance and related metabolic disorders.

## 6. Conclusions

Cancer and T2D present distinct metabolic shifts, with cancer exhibiting a predominantly glycolytic nature in contrast to the intricate metabolic profile of T2D. In cancer, various factors, such as mitochondrial dysfunction, low AMPK, elevated PDK levels, LDH, HIF-1α, decreased PDC levels, NADH, and mutations in oncogenes and tumor suppressor genes, along with influences from the tumor microenvironment, contribute to a pronounced bioenergetic shift known as the Warburg effect. The presence of one or more of these factors determines the tumor’s bioenergetic profile.

In the context of T2D, conflicting findings in the literature regarding OXPHOS status create challenges in clearly delineating an opposing metabolic shift between cancer and diabetes. Discrepancies emerge regarding the functionality of mitochondria and the activation of OXPHOS in various studies, a situation influenced by variations in experimental designs, examined tissues, employed methodologies, and possible misinterpretations, as argued by Wiseman et al. in 2019 [223]. Notably, in prediabetic conditions, there is evidence supporting an elevated insulin-induced OXPHOS status in response to persistent hyperglycemia. However, in T2D, OXPHOS exhibits either inactivity or activity in a tissue-specific manner, potentially linked to insulin resistance. Furthermore, T2D is characterized by heightened ROS levels, increased hepatic gluconeogenesis, and insulin resistance.

Targeted interventions designed to address the metabolic irregularities in both cancer and diabetes demonstrate promising outcomes in preclinical analyses, encompassing both in vivo and in vitro studies, as well as ongoing clinical trials. Further investigations into the efficacy and safety of potential nutrient adjuvants for patients with cancer and diabetes are much warranted.

## Figures and Tables

**Figure 1 biomedicines-12-00211-f001:**
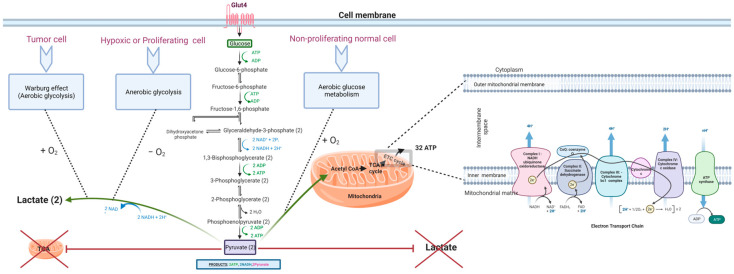
The glycolytic metabolic shift in tumor cells compared to the oxidative phosphorylation (OXPHOS) metabolic dominance in normal non-proliferating cells. (1) Pyruvate can enter the mitochondria to be converted to acetyl-CoA, which enters the Krebs cycle (tricarboxylic acid cycle or citric acid cycle). NADH and FADH_2_ produced from the Krebs cycle enter the ETC cycle to generate the energy molecule, adenosine triphosphate (ATP). OXPHOS, also called the ETC cycle, occurs in the inner membrane of the mitochondria and generates approximately 32 ATP energy molecules from a single glucose molecule. (2) The pyruvate generated in the cytoplasm from the breakdown of one glucose molecule is utilized by Warburg aerobic glycolysis or anaerobic glycolysis to produce lactate and a net of two ATPs. Abbreviations: ADP—adenosine diphosphate; ATP—adenosine triphosphate; ETC—electron transport chain; GLUT4—glucose transporter 4; NADH—nicotinamide adenine dinucleotide; O_2_—oxygen; TCA—tricarboxylic acid. Created with Biorender.com.

**Figure 2 biomedicines-12-00211-f002:**
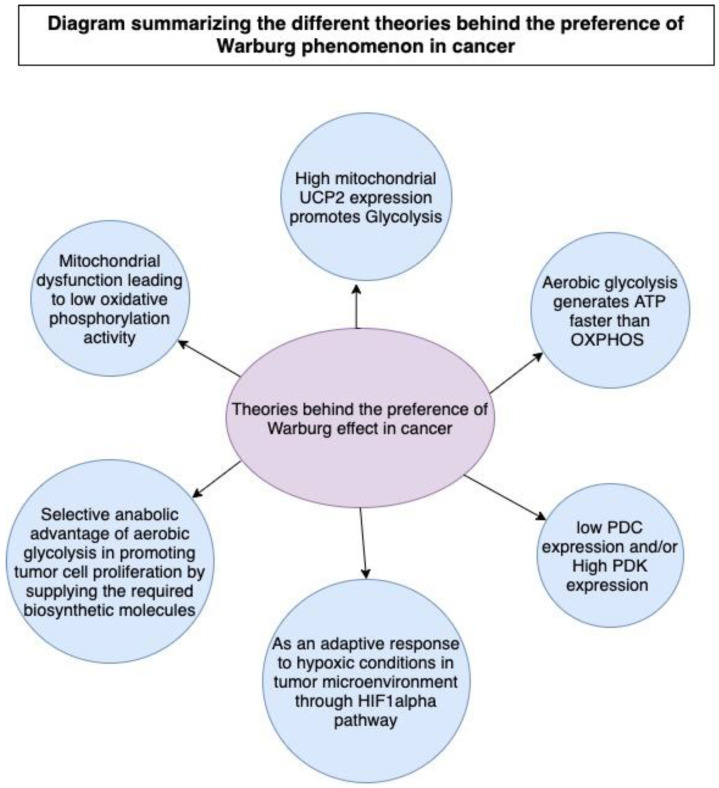
Different theories behind the Warburg effect in cancer. Abbreviations: ATP—adenosine triphosphate; HIF-1—hypoxia-inducible factor-1; OXPHOS—oxidative phosphorylation; PDC—pyruvate dehydrogenase complex; PDK—pyruvate dehydrogenase kinase; UCP—uncoupling protein 2. Created with Biorender.com.

**Figure 3 biomedicines-12-00211-f003:**
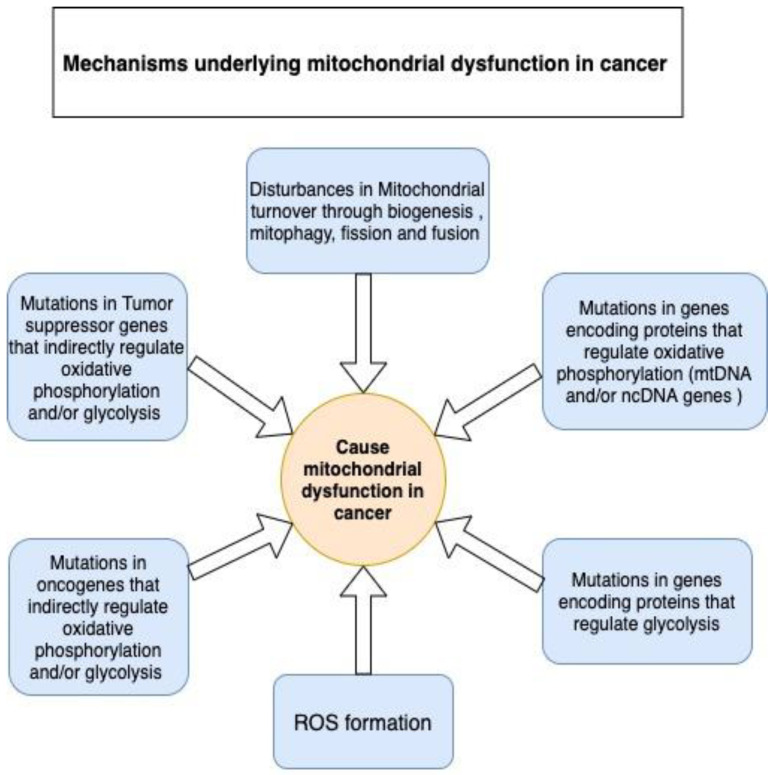
Mechanisms underlying mitochondrial metabolic dysfunction in cancer. Abbreviations: mtDNA—mitochondrial deoxyribonucleic acid; ncDNA—nuclear deoxyribonucleic acid; ROS—reactive oxygen species. Created with Biorender.com.

**Figure 4 biomedicines-12-00211-f004:**
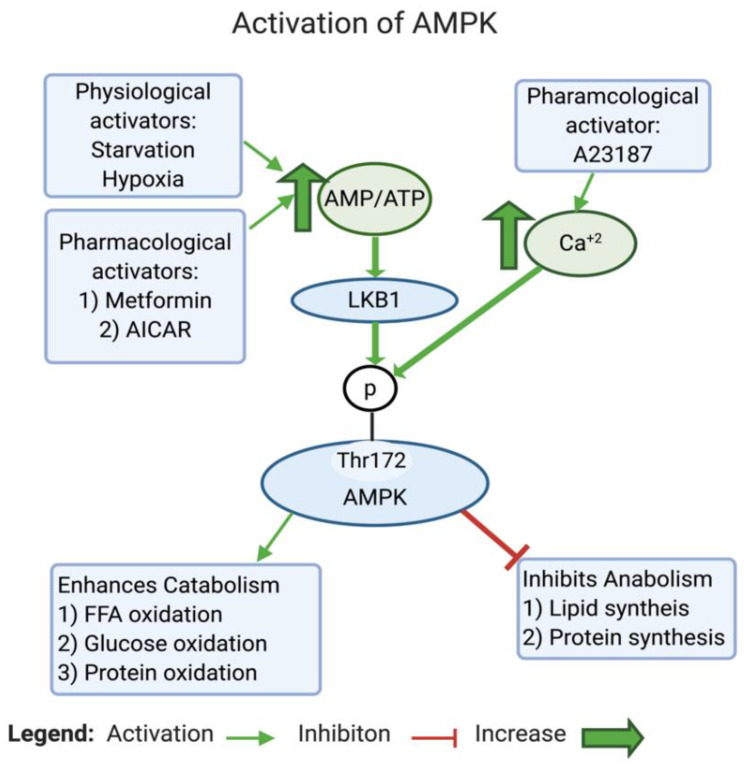
The activation of AMPK. Abbreviations: AICAR—5-aminoimidazole-4-carboxamide-1-β-D-ribofuranoside; AMP—adenosine monophosphate; AMPK—AMP-activated protein kinase; ATP—adenosine triphosphate; FFA—free fatty acids; LKB1—liver kinase B1; Thr172—threonine 172. Created with Biorender.com.

**Figure 5 biomedicines-12-00211-f005:**
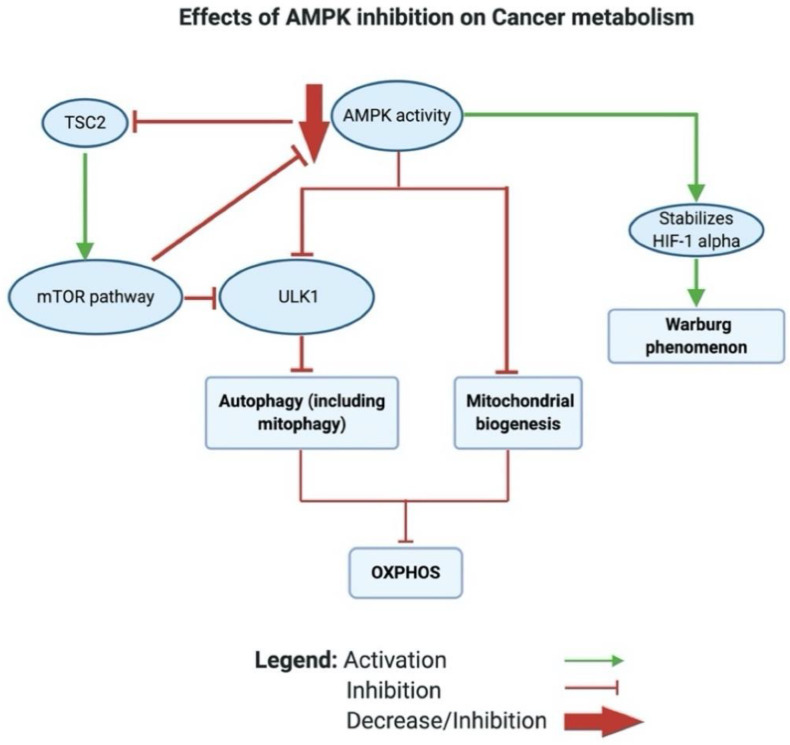
The effects of AMPK inhibition on the metabolic switch in cancer. The inhibition of AMPK promotes the shift towards the Warburg effect away from OXPHOS. Abbreviations: AMPK—AMP-activated protein kinase; HIF-1 alpha—hypoxia-inducible factor-1 alpha; mTOR—mammalian target of rapamycin; OXPHOS—oxidative phosphorylation; TSC2—tuberous sclerosis complex 2; ULK1—Unc-51 like autophagy activating kinase 1. Created with Biorender.com.

**Figure 6 biomedicines-12-00211-f006:**
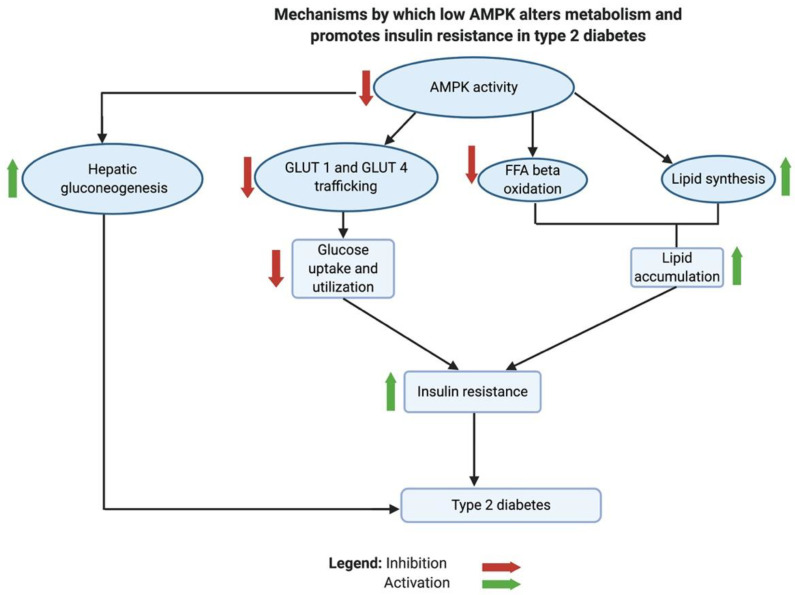
Effect of low AMPK activity in T2D. The main metabolic alterations are caused by low AMPK activity in T2D, which promotes insulin resistance. Abbreviations: AMPK—AMPK-activated protein kinase; FFA—free fatty acid; GLUT1—glucose transporter 1; GLUT4—glucose transporter 4. Created with Biorender.com.

**Table 4 biomedicines-12-00211-t004:** Summary of a group of flavonoids reported to reverse Warburg glycolysis towards mitochondrial respiration in preclinical studies.

Flavonoid Name	Flavonoid Subfamily	Mechanism of Targeting Warburg Glycolysis	Warburg Glycolytic Target	References
Apegenin	Flavones	Inhibited PKM2 activity and expression	PKM2	[351]
Epigallocatechin-3-gallate (EGCG)	Flavan-3-ols	Significant inhibition of PK activity and mRNA expression levels was observed at high concentrations. Inhibited HK2 enzymatic activity and reduced its protein levels. Decreased HIF-1a expression levels.	PKM2 HK2 HIF-1a	[352]
Proanthocyanidin B2 (PB2)	Anthocyanidins	Inhibited PKM2 enzyme through inhibition of its nuclear translocation and expression by interrupting interaction between PKM, HSP90 and HIF-1α	PKM2	[353]
Shikonin (SHI)	Naphthoquinone flavonoid	Repressed PKM2 activity	PKM2	[354,355]
Quercetin (QUE)	Flavonol	Suppressed PKM2 activity by regulating Akt-mTOR pathway	PKM2	[356]
		Inhibited HK2 by inhibiting Akt-mTOR pathway signaling	HK2	[356]
		Decreased levels of LDHA	LDH	[356]
Xanthohumol (XA)	Prenylated flavonoid	Suppressed HK2 activity by inhibiting EGFR-Ak signaling	HK2	[357]
10v	Synthetic flavonoid	Downregulated HK2	HK2	[358]
GL-V9	Synthetic flavonoid	Downregulated HK2 Detachment of HK2 from VDAC in the outer mitochondrial membrane induced apoptosis and inhibited glycolysis.	HK2	[359]
FV-429	Synthetic flavonoid	Detachment of HK2 from VDAC in the outer mitochondrial membrane induced apoptosis and inhibited glycolysis.	HK2	[360]
Gen-27	Synthetic flavonoid	Downregulated HK2 Detachment of HK2 from VDAC in the outer mitochondrial membrane induced apoptosis and inhibited glycolysis.	HK2	[361]
Astragalin (ASG)	O-glycoside flavonoid	Upregulated miR-125b expression, which reduced HK2 expression	HK2	[362]
Morin (MO)	Flavonol	Inhibited LDH activity	LDH	[363]
Methylalpinumisoflavon (MF)	Isoflavone	Suppressed HIF-1α activation	HIF-1α	[364]
Oroxylin A (OX-A)	Flavone	Destabilized HIF-1α through SIRT-3	HIF-1α	[365]
Baicalein (BA)	Flavone	Decreased HIF-1α expression	HIF-1α	[366]
Wogonin	O-methylated flavone	Suppression of HIF-1α by inhibiting PI3K/Akt pathway Induced phosphorylation and acetylation of P53 and inhibited MDM2 expression, which stabilized P53. P53 decreased the expression of key glycolytic enzymes.	HIF-1α	[367,368]
Berberine (BBR)	Isoquinoline flavonoid	Inhibit expression of HK2 by upregulating miR-145 Inhibited activity of PKM2 enzyme	HK2	[369,370]
Resveratrol		Deactivated HK2 by downregulating Akt signaling. Activated pyruvate dehydrogenase complex. Increased mitochondrial biogenesis and function.	HK2 PDH complex	[371,372,373]

## Data Availability

This review did not rely on any specific dataset. All referenced data are accounted for in the References section. The information presented in Appendix A was obtained from ClinicalTrials.gov on 23 November 2023, and any alterations made to the data are explicitly outlined in the legend in accordance with the ClinicalTrials.gov “Use of Data” policy.

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
