# Peer review of "Targeting the Metabolic Paradigms in Cancer and Diabetes"

_biomedicines, 2024, doi:10.3390/biomedicines12010211_

Round 1
Reviewer 1 Report (Previous Reviewer 1)
Comments and Suggestions for Authors
The authors present a complete review of how metabolic pathways interact in cancer and diabetes. It is easy to understand and follow, with appropriate figures that serve as support.
My general criticism is that I don't see how it differs from the numerous other similar articles that talk about the same. The authors should highlight what is new about this article and what makes it different.
Author Response
Thank you for your positive feedback. We have now clarified the distinctions of our article from others in the abstract. Your input is greatly appreciated.
Reviewer 2 Report (Previous Reviewer 2)
Comments and Suggestions for Authors
All concerns have been addressed and are ready for acceptance.
Author Response
Thank you so much for your positive feedback.
Reviewer 3 Report (New Reviewer)
Comments and Suggestions for Authors
The manuscript “Targeting the Metabolic Paradigms in Cancer and Diabetes” by Bosso and colleagues reviews the details of energy metabolism alterations in cancer and type 2 diabetes, and the therapeutic potential of some selected drugs and dietary supplements that could correct them.
The article is very long and it probably would have been easier for the reader to split it into two different reviews: one on the metabolic details about energy metabolism in cancer and diabetes, and one on the selected therapeutic and nutritional approaches. However, this is an editorial choice for the authors and the journal editors to make.
Here are my comments for the authors:
- The depiction of cancer and diabetes as two opposed mechanisms, to fit the yin-yang scheme, is a bit forced and leads to an oversimplification of what are really two completely different processes by focusing on just one - albeit important - metabolic perspective (energy metabolism). I recommend clarifying from the beginning that energy metabolism is just one of the many aspects of these diseases.
- For the same reason, the selected drugs and dietary supplements discussed here are just some among the many other therapeutic options that could be presented. Again, I would clarify that what is discussed here is just a selection of drugs and dietary supplement options, that focuses specifically on the energy metabolism part of the diseases.
- I recommend that the authors revise the abstract, which is not very easy to understand and a bit obscure especially in the first half. A sentence like the one you use in the conclusion section (“Cancer and type 2 diabetes are intricate metabolic disorders characterised by distinct yet seemingly opposite metabolic shifts”) would be a much more clear and effective way to introduce the topic.
- Line 11: “deregulated… metabolic alterations”. Something is wrong here, if they are alterations they are already deregulated
- Line 14, if you mention the Warburg effect in the abstract, you should briefly explain what it is
- When you introduce dietary supplements for cancer (Line 781) you state that two dietary compounds will be discussed (lipoic acid and flavonoids), but then (line 790) you say you will discuss four (including also omega-3 and artemisinin). However, these last two compounds are never discussed later in the review.
- As for diabetes, at Line 887 you say you will discuss two compounds (ALA and flavonoids), but then you discuss three (including glutamine). Please try to be more consistent throughout the text
- The section called “Discussion” that begins at Line 982 does not have a number, and it is misplaced here since it is not really a discussion of the previously presented material (about dietary supplements), but only a discussion of the two drugs metformin and TZD. It should be moved before section 5.
- The very last sentence (lines 1057-1059) is not relevant to what the article discussed (therapy, not prevention) and it does not really make sense: you cannot use a “corrective therapy” for prevention! I would remove this sentence.
-
Author Response
Thank you so much for your constructive comments. We have addressed your comments as follows;
- Regarding the length of the review article, we agree with you that it is long. However, it is common for such comprehensive reviews it be long. We hope that we made efforts to make the reading easy and connected logically. The reviewers' comments made this possible as well.
- Please see lines 904-906. We have mentioned only few dietary supplements that were shown to improve both cancer and diabetes for the sake of not making the review even longer.
- The Abstract has been paraphrased.
- Issues with lines 11 and 14 have been corrected.
- Omega3 and artemisinin were going to be included, then because we aimed at making the review shorter, we focused on Lipoic acid, flavonoids, and glutamine supplements only. Thanks for catching the mistake. We have removed Omega3 and Artemisinin errors.
- Line 901 (was Line 887) has been corrected to include glutamine.
- We have moved the previous discussion to a new section numbered 4.3 and titled :
4.3. Metformin and other AMPK-activators in Cancer Clinical Trials
- Thank you for this feedback, we have removed the last sentence of the conclusion.
Reviewer 4 Report (New Reviewer)
Comments and Suggestions for Authors
This review by Al-Mulla and co-workers is significantly summarized targeting the metabolic paradigms in cancer and diabetes exploring the metabolic pathways in cancer and T2D and highlighting promising nutrigenetic and therapeutic approaches to reverse the metabolic shift and restore a normal balance in both diseases. In this review, in first section explains the metabolic shifts in cancer disease and the reasons stepwise which is understandable to readers. The second section included a discussion about the metabolic shift in T2D and insulin resistance. In addition, various approaches in cancer and T2D have been summarized and discussed in the manuscript. Here, the author has made efforts to document and summarize common alterations reported in glycolysis-related genes in cancer and a group of flavonoids reported to reverse Warburg glycolysis towards mitochondrial respiration in preclinical studies which is useful to readers. Overall, the review article is informative, organized, and in good shape for the readers. Therefore, it can be accepted in "Biomedicine" publication. The cited references are fine.

Author Response
Many thanks for your positive feedback. We appreciate your comments on the review subject and overall organization. We are pleased that you found it useful and interesting.
This manuscript is a resubmission of an earlier submission. The following is a list of the peer review reports and author responses from that submission.
Round 1
Reviewer 1 Report
Comments and Suggestions for Authors
The presented review shows a comprehensive summary of how cell metabolism, especially glucose metabolism, affects cancer development and T2D. It also gives an excellent update on how metabolism hotspots can be used as therapeutic targets for treating both diseases.
However, I think some critical topics could be added to improve the article.
One of the most important aspects of the Warburg effect is the change of substrate as an energy source. In hypoxia or pseudo-hypoxia, cells look for alternatives to the use of glucose to maintain mitochondrial function, not so much for the synthesis of ATP that can be held only with anaerobic glycolysis, but by the need for metabolic intermediates such as aKG that are produced in TCA. One of the most studied alternatives is the use of glutamine by the mitochondria; I was surprised not to see something on this subject in your article. I think it would be a contribution to add something about glutamine metabolism and how it participates in both the development of cancer and T2D.
In addition, I find it fascinating how metformin is a common target for treating cancer and T2D. I think that it could be discussed a little at the conclusion level or in a separate paragraph how patients who consume metformin chronically due to T2D have a lower incidence of developing certain cancers, there are several studies in this regard in recent years, and since you focus part of your article in AMPK as a therapeutic target, it would be a contribution.
Does any AMPK activator have the same effect as metformin? It would also be interesting to discuss why metformin is so successful as a metabolic regulator compared to other drugs.
Are treatments using the PI3K/AKT pathway as a therapeutic target? I think you could argue a bit about this.
Thank you very much for your article; I hope you consider my remarks; I think your work has a lot of potential.
Reviewer 2 Report
Comments and Suggestions for Authors
A timely review article by Dr. Al-Mulla and the group elaborates on the role of altered metabolism and its therapeutic implications in cancer and diabetes. It is a very well-written document and worthwhile for the current translational aspect, though a few things need to be addressed before it is ready for acceptance. They are as follows:
1. In table 2, while discussing oncogenic KRAS's role in altered metabolism, authors must add a few more points under the pathways affected and other relevant columns. For further details, see PMID: 33870211 and PMID: 36256706.
2. Authors should also address the NRF2's role in chemoresistance and cancer metabolism, shown recently in PMID: 31911550 and PMID: 33922165.
3. While discussing the interconnection of AMPK and mTOR, authors should also point out the role of Phospholipase D (PLD), which regulates AMPK and mTOR in a feedback loop manner. Also, this opens up a therapeutic regimen to explore- combining rapamycin and metformin. This has been shown in PMID: 26323019 and PMID: 23895284.
4. Authors should add another table consisting of the relevant clinical trials discussed in this manuscript or which are currently ongoing with the relevance of the topic of this review.
